

# An improved hybrid whale optimization algorithm for global optimization and engineering design problems

Abolfazl Rahimnejad[1,*], Ebrahim Akbari[2,*], Seyedali Mirjalili[3,4], Stephen Andrew Gadsden[1], Pavel Trojovský[2] and Eva Trojovská[2]

[1] Department of Mechanical Engineering, McMaster University, Hamilton, Canada
[2] Department of Mathematics, University of Hradec Králové, Hradec Králové, Czech Republic
[3] Centre for Artificial Intelligence Research and Optimisation, Torrens University Australia, Adelaide, Australia
[4] Yonsei Frontier Lab, Yonsei University, Seoul, South Korea
[*] These authors contributed equally to this work.

Corresponding author
Abolfazl Rahimnejad,
a.rahimnejad@mcmaster.ca

## ABSTRACT

The whale optimization algorithm (WOA) is a widely used metaheuristic optimization approach with applications in various scientific and industrial domains. However, WOA has a limitation of relying solely on the best solution to guide the population in subsequent iterations, overlooking the valuable information embedded in other candidate solutions. To address this limitation, we propose a novel and improved variant called Pbest-guided differential WOA (PDWOA). PDWOA combines the strengths of WOA, particle swarm optimizer (PSO), and differential evolution (DE) algorithms to overcome these shortcomings. In this study, we conduct a comprehensive evaluation of the proposed PDWOA algorithm on both benchmark and real-world optimization problems. The benchmark tests comprise 30-dimensional functions from CEC 2014 Test Functions, while the real-world problems include pressure vessel optimal design, tension/compression spring optimal design, and welded beam optimal design. We present the simulation results, including the outcomes of non-parametric statistical tests including the Wilcoxon signed-rank test and the Friedman test, which validate the performance improvements achieved by PDWOA over other algorithms. The results of our evaluation demonstrate the superiority of PDWOA compared to recent methods, including the original WOA. These findings provide valuable insights into the effectiveness of the proposed hybrid WOA algorithm. Furthermore, we offer recommendations for future research to further enhance its performance and open new avenues for exploration in the field of optimization algorithms. The MATLAB Codes of FISA are publicly available at https://github.com/ebrahimakbary/PDWOA.

## INTRODUCTION

As optimization problems in various disciplines become increasingly challenging, it becomes apparent that classical optimization methods suffer from limitations. These limitations include convergence to local optima, requirements of differentiability and

continuity, and high computational burdens. Consequently, there is a growing need to develop more robust tools for optimal problem-solving. In recent years, metaheuristic methods, such as particle swarm optimization (PSO) (*Kennedy & Eberhart, 1995*) and genetic algorithm (GA) (*Holland, 1992*), have gained popularity and success in solving optimization problems. Various metaheuristic methods are still being proposed such as the termite life cycle optimizer (TLCO) (*Minh et al., 2023b*; *Minh et al., 2023a*), K-means optimizer (KO) (*Minh et al., 2022*), planet optimization algorithm (POA) (*Sang-To et al., 2022*), a combination of artificial neural network (ANN) and balancing composite motion optimization (BCMO) (*Tran et al., 2023*), and the new movement strategy of cuckoo search (NMS-CS) (*Cuong-Le et al., 2021*).

Researchers tend to utilize metaheuristic methods for optimization problems due to their derivative-free formulation and their ability to escape local optima and find global optima. However, it is important to consider the No Free Lunch theorem (*Wolpert & Macready, 1997*), which suggests that no single optimization algorithm performs best for all problems. Therefore, there is a need to explore and develop new metaheuristic algorithms that are specifically designed to address the challenges of different optimization problems.

The whale optimization algorithm (WOA) is a recent metaheuristic method suggested by *Mirjalili & Lewis (2016)*, inspired by the hunting strategy of humpback whales. WOA has gained significant attention from engineers, designers, and researchers worldwide for its effectiveness in optimizing various problems. However, the original WOA formulation has a limitation: it only considers the best solution from each iteration, neglecting valuable information from other individuals and their best positions. This limitation can hinder the algorithm's overall optimization performance.

To address this drawback, our proposed approach introduces an enhanced version of WOA called the Pbest-guided differential Whale Optimization Algorithm (PDWOA). PDWOA incorporates efficient features from PSO and differential evolution (DE) algorithms (*Storn & Price, 1997*) to improve the algorithm's ability to avoid local optima and achieve global optima, particularly in shifted optimization problems. In addition, two non-parametric statistical tests, including the Wilcoxon signed-rank test and the Friedman test (*Derrac et al., 2011*; *Buch, Trivedi & Jangir, 2017*; *Ghasemi et al., 2023*), are employed to validate the performance improvements achieved by PDWOA over the original WOA.

The contributions of this study are outlined as follows:

1. Overview and analysis of the Whale Optimization Algorithm (WOA) to understand its functionality and limitations, particularly in complex real-world problems.
2. Development of a new enhanced version of WOA known as the Pbest-guided differential Whale Optimization Algorithm (PDWOA) to address the identified limitations of the original algorithm.
3. Evaluation of the performance of PDWOA compared to the original WOA through experiments on 30 shifted test functions from CEC2014. The results demonstrate the efficiency of PDWOA in obtaining optimal solutions. Statistical tests, such as the Wilcoxon signed-rank test and the Friedman test, are employed to validate the performance improvements.

4. Application of PDWOA to solve three real-world engineering problems, providing practical validation of its optimization performance in real-world scenarios.

5. Discussion of potential future improvements by exploring the integration of models from other powerful optimization algorithms, aiming to expand the range of problems that can be accurately solved by the proposed algorithm.

The remaining sections of this paper are organized as follows. The "Related Work" section provides an overview of the related work in the field. "WOA" presents a brief introduction to the WOA. The "Challenges and Enhanced Hybrid Version of WOA" section discusses the main drawbacks of WOA and proposes the Pbest-guided differential WOA (PDWOA) by incorporating efficient features of PSO and DE algorithms. The "Simulation Results" section presents the simulation results, where extensive experiments are conducted to evaluate the performance of PDWOA, including the statistical tests. "Discussion and Future Studies" discusses the results and provides potential areas for future studies. Finally, the paper is concluded in the "Conclusion" section.

## RELATED WORKS

A comprehensive overview of the applications of WOA, including various improvements, has been presented in *Gharehchopogh & Gholizadeh (2019)*. Some notable examples of these improvements include the use of WOA for detecting weak signals in rotating (*He et al., 2019*), analyzing clinical data of anaemic pregnant (*Saidala & Devarakonda, 2017*), scheduling tasks in cloud computing (*Sreenu & Sreelatha, 2017*), and suppressing sidelobe in multiple input and multiple output radar systems (*Yuan et al., 2018*). Additionally, *Mohammadi & Mehdizadeh (2020)* proposed a novel hybrid model that combines support vector regression with WOA for the daily estimation of reference evapotranspiration, demonstrating superior performance compared to support vector regression-only models.

*Qais, Hasanien & Alghuwainem (2020a)* proposed a new enhanced version of WOA, called EWOA, specifically designed for maximizing power extraction from variable-speed wind generators (VSWGs). Instead of using the parameters suggested in the original WOA, EWOA incorporates a cosine function to control the searching and encircling behavior. *Wang & Chen (2020)* proposed a novel approach for medical diagnosis by improving a support vector machine (SVM) using chaotic WOA with multiple swarms (CMWOA). Their technique exhibited excellent performance in terms of avoiding local optima and achieving fast convergence. *Cao et al. (2020)* incorporated chaos theory to enhance the exploration ability and convergence characteristics of WOA, resulting in the development of a new chaotic-based improved version called CIWOA. This approach was specifically applied to achieve efficient terminal voltage control for proton exchange membrane fuel cells (PEMFCs).

*Akyol & Alatas (2020)* applied WOA and social impact theory based optimization for sentiment classification in online social media. Furthermore, *Zeng et al. (2021)* proposed a competitive mechanism enhanced WOA (CMWOA) for effectively addressing multi-objective optimization problems. *Qais, Hasanien & Alghuwainem (2020b)* introduced a novel design of Sugeno fuzzy logic controllers (FLCs) based on WOA (WOA-FLCs) to

enhance the low voltage ride-through of VSWGs, resulting in improved time response characteristics surpassing those obtained by GA and grey wolf optimizer (GWO). *Jain, Katarya & Sachdeva (2020)* employed a novel social network-based WOA (SNWOA) to identify opinion leaders in social networks. *Rosyadi, Penangsang & Soeprijanto (2017)* applied the WOA to determine the optimal placement and size of filters in distribution systems. *Chen, Li & Yang (2020)* utilized chaos mechanism and quasi-opposition to enhance the convergence speed of WOA and mitigate the issue of local optima when solving large-scale problems. *Liu et al. (2020)* proposed the utilization of WOA for evaluating the resilience of regional flood disasters, demonstrating improved generalization performance and remarkable stability. *Wang et al. (2019)* introduced an opposition-based variant of WOA for tackling multi-objective optimization problems.

*Srivastava et al. (2018)* utilized WOA to estimate the parameters of a permanent magnet synchronous motor. An improved version of WOA optimizer was suggested in *Abdel-Basset, Mohamed & Mirjalili (2021)*, which comprises three modifications compared to the original WOA. Firstly, the dynamic distance control factor was used rather than a fixed one. Secondly, a certain probability was used to achieve a compromise between movement towards the best solution and its opposite for escaping from local optimal solutions. Finally, Nelder–Mead was used along with the Pareto archived evolution strategy (PAES) to further improve WOA. Authors of *Mahdad (2018)* solved the optimal power flow (OPF) problem utilizing a new partitioning whale algorithm.

In *Chen et al. (2020)*, an improved WOA named RDWOA was suggested for improving the convergence and global optimization performance of WOA in solving multi-dimensional problems. The improvement included two schemes, random spare or random replacement and double adaptive weight, which were used for advancing the convergence, exploration at the initial phases, and exploitation at subsequent phases. The proposed strategies considerably increased the convergence speed and the optimization performance of WOA. The efficiency of RDWOA was proved by utilizing typical benchmarks and engineering problems.

*Trivedi et al. (2016)* applied WOA to solve emission constraint environmental economic dispatch problems. *Tu et al. (2021)* proposed another enhanced variant of WOA to improve its convergence performance and prevent being trapped in local optimal solutions. The enhancement employs a new communication mechanism (CM) for improving the global optimization performance and biogeography-based optimization (BBO) to compromise between the exploring and exploiting performances. The effectiveness of BBO was confirmed using benchmark and engineering problems.

*Abdel-Basset, Abdle-Fatah & Sangaiah (2018)* proposed an enhanced version of WOA that incorporates Lévy flight (LF) for problem-solving in the cloud computing environment. *Mafarja & Mirjalili (2017)* proposed a hybrid approach that combines WOA with simulated annealing for feature selection. In *Nazari-Heris et al. (2017)*, the optimal generations of combined heat and power units were determined using WOA. *Guo et al. (2020)* augmented WOA by incorporating adaptive social learning (ASL) and wavelet mutation. At first, a novel exploration probability was formulated for improving the performance of WOA. Then, an ASL strategy was utilized for constructing the adaptive social network (ASN)

of the WOA population, to enhance its diverseness. Finally, the suggested procedure was augmented using the Morlet wavelet mutation strategy. WOA was proposed in *Reddy, Reddy & Manohar (2017)* for the solution of optimally identifying the size of renewable energy resources.

In *Samadianfard et al. (2020)* a hybridization of the multi-layer perceptron (MLP) neural network and WOA was roposed for wind speed forecasting. Content-based image retrieval was solved in *Aziz, Ewees & Hassanien (2018)* using multi-objective WOA (MOWOA) algorithm. In *Wu et al. (2018)*, the path planning problem for solar-powered UAV in urban environment was solved using WOA enhanced with adaptive chaos-Gaussian switching solving strategy and coordinated decision-making mechanism. In *Hou et al. (2020)*, a hybrid of quantum simultaneous WOA (QSWOA) and a multi-objective economic model predictive control (MOEMPC) was proposed for controlling gas turbines. A new improved opposition-based WOA (IOWOA) was used for estimating the parameters of solar cells diode models (*Abd Elaziz & Oliva, 2018*). Binary WOA was utilized in *Eid (2018)* to deal with feature selection problems. In *Liu, Yao & Li (2020)* a hybridization of LF-augmented WOA and DE was suggested for dealing with the job shop scheduling problem (JSSP), where, LF and DE are used for improving the exploration and exploitation performances, respectively. Data clustering based on WOA was proposed in *Canayaz & Özdağ (2017)*. *Qiao et al. (2020)* employed a novel improved variant of WOA called IWOA for short-term natural gas consumption forecasting. *Khalilpourazari, Pasandideh & Ghodratnama (2018)* proposed the utilization of Whale Optimization Algorithm (WOA) and Water Cycle Algorithm (WCA) for programming a multi-item economic order quantity model. *Pham et al. (2020)* proposed the utilization of WOA for the optimal allocation of resources in wireless networks.

## WOA

The mathematical model of the original WOA, inspired by the hunting strategy of humpback whales, which consists of the stages of surrounding prey, bubble-net hunting maneuver, and search for prey, is briefly discussed in this section, *Mirjalili & Lewis (2016)*.

### Encircling prey

During the encircling prey stage, the WOA algorithm emulates the ability of humpback whales to recognize the prey's position and encircle them. In each iteration, the best solution, acting as the leader, is considered the target prey. The behavior is defined in Eqs. (1)–(3):

$$\vec{X}(t+1) = \vec{X}_{Leader}(t) - \vec{A} \odot \vec{D} \tag{1}$$

$$\vec{A} = 2a.\vec{r} - a \tag{2}$$

$$\vec{C} = 2.\vec{r} \tag{3}$$

where, $t$ represents the iteration number, $\vec{A}$ and $\vec{C}$ are the coefficient vectors of WOA, $\vec{X}_{Leader}$ denotes the position vector of the best solution found so far, and $\vec{X}$ represents the position vector of each member in the algorithm population. Furthermore, the $\odot$ sign denotes the element-wise multiplication. Notably, to enhance the performance of WOA, the value of $a$ linearly declines from 2 to zero during the iterations, and $\vec{r}$ is a vector of uniformly distributed random numbers between zero and one.

**Exploitation stage: bubble-net attacking maneuver**

The bubble-net attacking maneuver, inspired by the hunting behavior of humpback whales, is modeled using two strategies:

**1. Declining surrounding strategy:** This strategy is achieved by reducing the value of $a$ in Eq. (2). Notably, the range of variation in vector $\vec{A}$ is directly proportional to $a$, where $\vec{A}$ consists of randomly generated values between $-a$ and $a$.

**2. Spiral position update:** The whale's displacement towards the prey's position, simulating the spiral motion of humpback whales, is formulated as Eq. (4):

$$\vec{X}(t+1) = \vec{X}_{Leader}(t) + \vec{D'}.e^{B.L}.Cos(2\pi L) \tag{4}$$

where $\vec{D'} = \left| \vec{X}_{Leader} - \vec{X} \right|$ denotes how far is the $i$th whale from the prey, $B$ is a constant that describes the logarithmic spiral motion, and $L$ is a random value between $-1$ and $1$.

It is important to mention that the selection between the declining surrounding strategy and the spiral position update is equally probable.

**Search for prey**

The search for prey, representing the exploration stage of WOA, can be achieved by adjusting the vector $\vec{A}$. This mechanism facilitates a global search by setting the absolute value of the vector $\vec{A}$ to $|\vec{A}| > 1$. Mathematically, this stage can be formulated as Eqs. (5) and (6):

$$\vec{D} = \left| \vec{C} \odot \vec{X}_{rand} - \vec{X} \right| \tag{5}$$

$$\vec{X}(t+1) = \vec{X}_{rand} - \vec{A} \odot \vec{D} \tag{6}$$

where $\vec{X}_{rand}$ denotes the position vector of an indiscriminately chosen solution. It is important to note that the WOA method relies on two main parameters, $\vec{A}$ and $\vec{C}$, which need to be tuned.

## CHALLENGES AND ENHANCED HYBRID VERSION OF WOA

In the current section, the challenges of WOA are discussed, followed by the introduction of a novel improved version of WOA to address those challenges.

## Challenges of WOA

In practical applications, we encounter optimization problems with diverse behaviors and levels of complexity. Therefore, researchers strive to find an algorithm that is robust, requires minimal parameter tuning, and offers simplicity and fast convergence speed (*Talbi, 2009*). Real-world problems often involve shifted functions, where the global optimal solutions do not reside at the origin of coordinates and vary across dimensions. It is well-documented in the literature that many algorithms exhibit reduced performance for shifted functions (*Liang, Qu & Suganthan, 2013*), which necessitates appropriate modifications.

To investigate this issue with WOA, we conducted experiments using the conventional model of the sphere function (*Mirjalili & Lewis, 2016*) and its shifted counterpart, known as the Shifted Sphere Function (*Suganthan et al., 2005*). We aimed to determine the optimal solutions for these functions with 30 dimensions using WOA, PSO, and DE methods. Each function was independently evaluated in 25 runs, with 300,000 function evaluations (*Suganthan et al., 2005*) and a population size of 30 for the algorithm. The mean values obtained by WOA for the optimal response of the traditional sphere function and the shifted sphere function were 0 and 0.478, respectively. Figure 1 illustrates the convergence characteristics of WOA, DE/best/1, and PSO algorithms for both functions. It is worth noting that all algorithm parameters were set according to the recommendations in the original codes, leading to improved average performance across a wide range of problems. From the figure, it is evident that the original WOA exhibits reduced performance for shifted functions. Therefore, it is crucial to either tune the key controlling parameters or modify the WOA formulation to enhance its efficiency in solving a wider range of engineering and real-world problems.

Another issue with the original WOA is that it only stores the best solution among the entire population in each iteration. In contrast, algorithms like particle swarm optimization (PSO) store the personal best position (Pbest) for each member in each iteration, which enables directing the population members to avoid local optima. Therefore, an enhanced hybrid model of WOA can be developed by leveraging the advantageous features of other algorithms as an auxiliary operator. In this study, we present a new efficient hybrid variant of WOA that incorporates the formulations of PSO (*Eberhart & Kennedy, 1995*) and differential evolution (DE) (*Storn & Price, 1997*). This hybrid variant will be discussed in detail in the next section.

## Pbest-guided differential WOA

The storage of only the best solution in WOA, similar to GA, is identified as a fundamental weakness of the algorithm based on our investigation. This limitation arises from eliminating many candidate solutions in each iteration, which could potentially be useful in subsequent iterations and enhance the algorithm's optimization capability, as observed in DE and PSO algorithms. Consequently, we can leverage the models/formulations of basic DE, PSO, and their advanced variants, which have gained significant popularity in recent years, to enhance WOA's performance in locating the global optimum of real-world optimization problems.

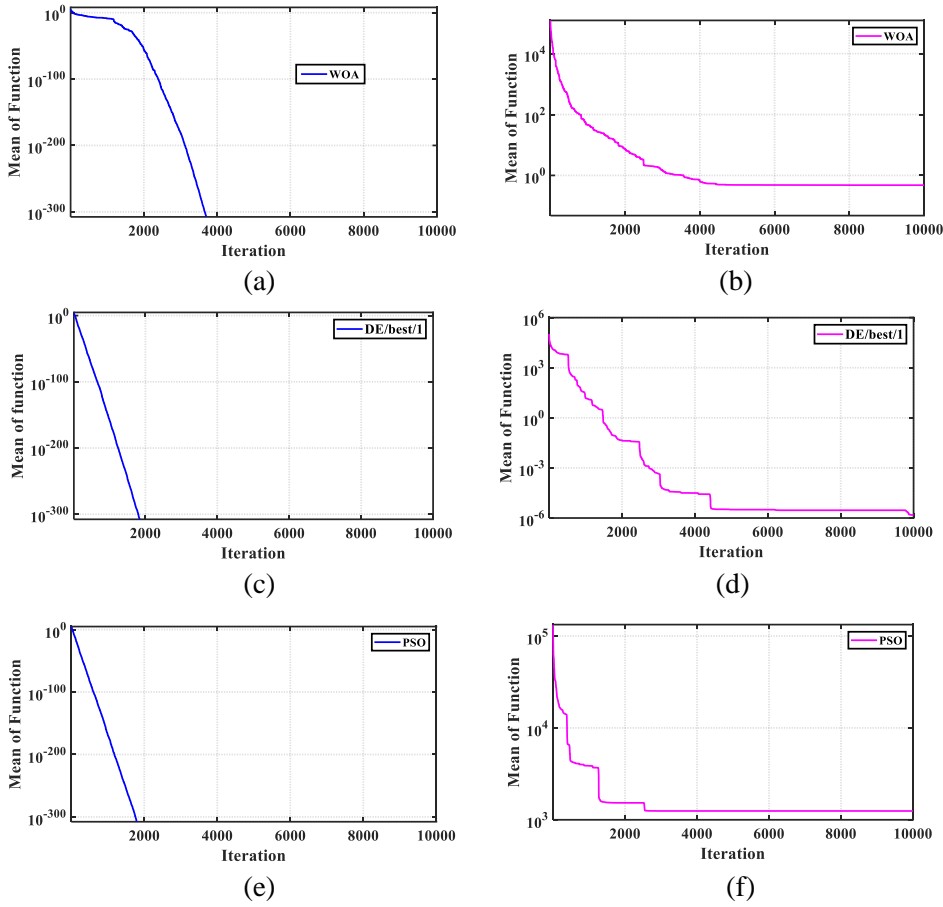

**Figure 1** **The convergence characteristics for the standard test functions.** (A) Sphere function solution obtained by WOA; (B) shifted sphere function solution obtained by WOA (mean is 4.786e−1); (C) sphere function solution obtained by DE/best/1; (D) shifted sphere function solution obtained by DE/best/1 (mean is 1.510e−06); (E) sphere function solution obtained by PSO; (F) shifted sphere function solution obtained by PSO (mean is 1.247e3).

*PSO-based Modification:* The first modification proposed in this study involves storing the personal best(*Pbest*) position of each member in each iteration, denoted by $\rightarrow Xpbest > 1$, similar to the PSO algorithm. With this Pbest-guided modification, the search equations can be rewritten as Eqs. (7)–(14):

$$\vec{A} = 2a.\vec{r} - a \tag{7}$$

$$\vec{C} = 2\vec{r} \tag{8}$$

$$\vec{D} = \left| \vec{C} \odot \vec{X}_{Leader}(t) - X\vec{pbest}(t) \right| \tag{9}$$

$$\vec{X}(t+1) = \vec{X}_{Leader}(t) - \vec{A} \odot \vec{D} \tag{10}$$

$$\vec{D}'' = \left| \vec{C} \odot \vec{X}_{rand} - \vec{Xpbest} \right| \tag{11}$$

$$\vec{X}(t+1) = \vec{X}_{rand} - \vec{A} \odot \vec{D}'' \tag{12}$$

$$\vec{D}' = \left| \vec{X}_{Leader} - \vec{Xpbest} \right| \tag{13}$$

$$\vec{X}(t+1) = \vec{X}_{Leader}(t) + \vec{D}'.e^{BL}.Cos(2\pi L) \tag{14}$$

where Eqs. (9) and (10) denote the encircling prey phase, Eqs. (11) and (12) model the search for prey phase, and Eqs. (13) and (14) demonstrate the spiral position update. Note that in the proposed algorithm, the same as PSO, for each member of the population in each iteration, $\vec{Xpbest}$ is updated for each individual as Eq. (15):

$$\vec{Xpbest}(t+1) = \begin{cases} \vec{X}(t+1); if f\left(\vec{X}(t+1)\right) \leq f\left(\vec{Xpbest}(t)\right) \\ \vec{Xpbest}(t); else \end{cases} \tag{15}$$

***DE-based modification:*** In the DE-based modification, we incorporate the best position found by each individual in all previous iterations. This enables us to leverage the mutations proposed in the DE algorithm to effectively enhance the original WOA. Therefore, in the second stage of the modification, a mutation phase, as defined in Eq. (16), is added to the formulation of WOA immediately after the main phases of the algorithm:

$$\vec{V}(t) = \vec{Xpbest}(t) + rand1\left(\vec{X}_{Leader}(t) - \vec{Xpbest}(t)\right) \\ + rand2\left(\vec{Xpbest}_{r1}(t) - \vec{Xpbest}_{r2}(t)\right) \tag{16}$$

where $\vec{Xpbest}_{r1}$ and $\vec{Xpbest}_{r2}$ are the personal best positions of two solutions randomly chosen from the population for updating each solution. Similarly, $rand1$ and $rand2$ are random vectors with dimensions equal to $D$ (problem's dimension), where the elements' values range between 0 and 1. Subsequently, a random variable $randj$ is generated for each dimension $j$ of each solution, leading to Eq. (17):

$$\vec{X_j}(t+1) = \begin{cases} \vec{V_j}(t) & if \ randj > Cr \\ \vec{X_j}(t) & otherwise \end{cases} . \tag{17}$$

Here, *Cr* represents a control parameter, similar to the crossover rate used in evolutionary algorithms.

Finally, we want to emphasize that we employed the penalty method, a widely-used approach for addressing constraints in constrained optimization problems, in our research. The penalty method utilizes penalty functions to guide the optimization algorithm towards feasible solutions while penalizing infeasible solutions. To provide a visual representation of the proposed approach, we have included a flowchart of the Pbest-guided differential Whale Optimization Algorithm (PDWOA) in Fig. 2.

## SIMULATION RESULTS

The verification of the effectiveness of PDWOA is achieved using two sets of experiments, firstly it was used for solving CEC 2014 Test Functions (*Liang, Qu & Suganthan, 2013*), and then it is exploited for solving three engineering problems.

### Solving CEC 2014 test functions using PDWOA

In order to compare the performance of PDWOA with that of the original WOA, 30 test functions with 30 dimensions have been selected from CEC 2014 Test Functions (*Liang, Qu & Suganthan, 2013*). These functions include unimodal (F1-F3), multimodal (F4-F16), hybrid (F17-F22), and composition (F23-F30) functions. For both algorithms, we consider a population number equal to 30 and iteration numbers equal to 10,000, *i.e.,* the number of function evaluations (NFEs) done by each algorithm (for each test function) is 300,000. To find the optimal solution of each function, 25 separate runs have been executed for each algorithm and then, statistical analysis has been performed on the results.

A comparative study between DE, PSO, the original WOA, and the proposed PDWOA with three different *Cr* settings, *i.e.,* a random value and the fixed values of 0.1 and 0.9, is presented in Table 1. In this table, the terms "Mean" and "Std". represent the average value and standard deviation, respectively, of the results obtained from 25 independent runs for optimizing each function using each algorithm. The term "Rank" indicates the ranking of the algorithm's Mean index, reflecting its effectiveness in optimizing the considered function. Additionally, "NB" represents the number of functions for which the algorithm achieves the best Mean index, while "MR" represents the mean of the Rank indices of the algorithm across all functions. It is evident from the table that the proposed algorithms, with two different *Cr* tunings, outperform the original algorithm significantly. Specifically, the PDWOA with *Cr* set to a random value and 0.1 surpasses the performance of the original WOA for 21 and 24 shifted test functions, respectively. Notably, even in cases where the suggested algorithm exhibits worse performance, the resulting outcomes do not deviate significantly from those obtained by the original WOA.

It can be further seen from results in Table 1 the suggested PDWOA could attain results of a much higher quality for test functions F1, F2, F3, F7, F10, F17, F18, F20, and F30 compared to the original algorithm. Furthermore, the convergence characteristics of the

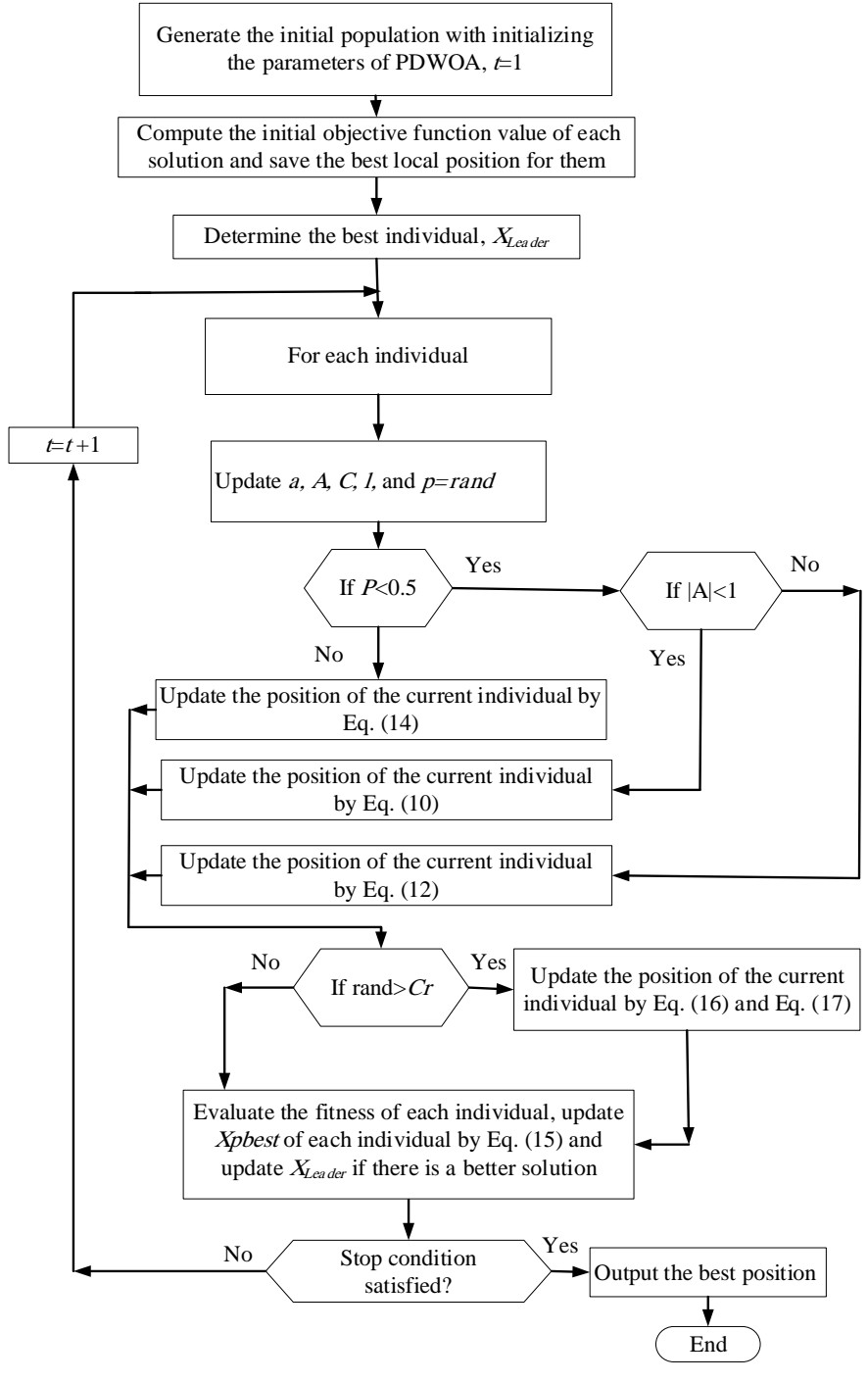

**Figure 2   Flowchart of PDWOA.**

algorithms for some of the test functions are depicted in Fig. 3, which confirms the higher performance and convergence rate of the suggested PDWOA.

Table 2 presents the average simulation time of 25 runs for each of the CEC 2014 test functions, with the aim of comparing the computational burden of the proposed PDWOA

**Table 1 Summary of the results of DE, PSO, WOA and different variants of PDWOA for CEC 2014 test functions.**

| Function | DE/best/1 Mean Std. Rank | PSO Mean Std. Rank | WOA Mean Std. Rank | PDWOA/$Cr$ = rand Mean Std. Rank | PDWOA/$Cr$ = 0.1 Mean Std. Rank | PDWOA/$Cr$ = 0.9 Mean Std. Rank |
|---|---|---|---|---|---|---|
| F1 | 1.11E+08 4.23E+07 6 | 4.38E+07 1.81E+07 5 | 3.39E+07 1.96E+07 4 | 3.15E+06 1.75E+06 1 | 3.31E+06 2.11E+06 2 | 4.08E+06 2.92E+06 3 |
| F2 | 1.28E+10 6.28E+09 6 | 9.83E+08 5.36E+08 5 | 3.43E+06 1.38E+06 4 | 1.42E+04 1.32E+04 1 | 2.45E+04 1.43E+04 3 | 1.46E+04 1.26E+04 2 |
| F3 | 9.32E+04 5.80E+04 6 | 2.84E+04 1.40E+04 4 | 4.50E+04 3.03E+04 5 | 3.51E+03 3.56E+03 2 | 1.59E+03 2.11E+03 1 | 1.13E+04 1.85E+04 3 |
| F4 | 1.27E+03 6.92E+02 6 | 2.61E+02 8.36E+01 5 | 1.97E+02 4.93E+01 4 | 1.23E+02 3.19E+01 2 | 1.03E+02 4.01E+01 1 | 1.37E+02 4.45E+01 3 |
| F5 | 2.09E+01 5.00E−02 6 | 2.04E+01 6.00E−02 5 | 2.02E+01 1.70E−01 3 | 2.02E+01 1.50E−01 2 | 2.03E+01 2.70E−01 4 | 2.02E+01 3.30E−01 1 |
| F6 | 2.04E+01 3.53E+00 1 | 2.95E+01 2.70E+00 4 | 3.71E+01 4.11E+00 5 | 2.63E+01 3.33E+00 2 | 3.45E+01 4.44E+00 3 | 4.00E+01 1.86E+00 6 |
| F7 | 1.33E+02 7.20E+01 6 | 1.50E+01 1.36E+01 5 | 1.03E+00 4.00E−02 4 | 9.00E−02 1.00E−01 1 | 1.80E−01 3.30E−01 3 | 1.00E−01 1.20E−01 2 |
| F8 | 1.08E+02 3.04E+01 2 | 1.46E+02 2.85E+01 3 | 1.79E+02 2.34E+01 4 | 6.64E+01 1.68E+01 1 | 1.85E+02 2.82E+01 5 | 2.15E+02 6.27E+01 6 |
| F9 | 1.76E+02 4.24E+01 1 | 1.78E+02 2.72E+01 2 | 2.29E+02 2.83E+01 5 | 2.05E+02 5.24E+01 3 | 2.22E+02 6.47E+01 4 | 2.38E+02 4.51E+01 6 |
| F10 | 3.07E+03 4.54E+02 2 | 4.37E+03 6.39E+02 6 | 3.94E+03 1.05E+03 4 | 6.27E+02 4.05E+02 1 | 3.54E+03 7.29E+02 3 | 4.36E+03 4.04E+02 5 |
| F11 | 3.15E+03 6.56E+02 1 | 4.90E+03 9.12E+02 6 | 4.77E+03 8.17E+02 5 | 4.11E+03 8.53E+02 2 | 4.48E+03 5.99E+02 4 | 4.47E+03 1.11E+03 3 |
| F12 | 2.13E+00 1.06E+00 6 | 1.36E+00 3.90E−01 2 | 1.42E+00 6.10E−01 4 | 1.15E+00 6.70E−01 1 | 1.39E+00 6.60E−01 3 | 1.57E+00 3.80E−01 5 |
| F13 | 3.03E+00 9.40E−01 6 | 5.40E−01 7.00E−02 3 | 5.20E−01 8.00E−02 2 | 5.10E−01 1.30E−01 1 | 6.30E−01 1.40E−01 5 | 5.60E−01 1.20E−01 4 |
| F14 | 4.99E+01 3.52E+01 6 | 1.63E+00 2.84E+00 5 | 2.70E−01 4.00E−02 2 | 3.00E−01 5.00E−02 3 | 2.60E−01 5.00E−02 1 | 4.10E−01 3.30E−01 4 |
| F15 | 7.41E+04 7.77E+04 6 | 1.05E+02 6.57E+01 3 | 7.88E+01 1.87E+01 2 | 4.55E+01 1.21E+01 1 | 5.26E+02 6.32E+02 4 | 7.92E+02 2.71E+02 5 |

**Table 1** (*continued*)

| Function | DE/best/1 Mean Std. Rank | PSO Mean Std. Rank | WOA Mean Std. Rank | PDWOA/$Cr$ = rand Mean Std. Rank | PDWOA/$Cr$ = 0.1 Mean Std. Rank | PDWOA/$Cr$ = 0.9 Mean Std. Rank |
|---|---|---|---|---|---|---|
| F16 | 1.15E+01 4.00E−01 1 | 1.28E+01 5.60E−01 4 | 1.24E+01 4.60E−01 3 | 1.17E+01 4.00E−01 2 | 1.31E+01 3.30E−01 6 | 1.30E+01 5.80E−01 5 |
| F17 | 3.65E+06 2.86E+06 5 | 1.14E+06 8.15E+05 4 | 5.02E+06 2.18E+06 6 | 7.82E+05 3.34E+05 3 | 6.39E+05 6.53E+05 1 | 7.74E+05 2.98E+05 2 |
| F18 | 1.17E+06 2.57E+06 6 | 5.40E+03 2.45E+03 4 | 3.32E+04 7.93E+04 5 | 4.34E+03 4.79E+03 1 | 4.79E+03 6.86E+03 2 | 4.80E+03 6.01E+03 3 |
| F19 | 7.42E+01 4.08E+01 5 | 3.23E+01 2.50E+01 2 | 5.66E+01 5.04E+01 3 | 2.39E+01 2.67E+01 1 | 6.72E+01 5.07E+01 4 | 8.91E+01 3.61E+01 6 |
| F20 | 2.71E+04 3.00E+04 5 | 1.04E+04 5.33E+03 4 | 2.90E+04 2.30E+04 6 | 7.76E+03 4.60E+03 3 | 2.33E+03 1.82E+03 1 | 3.56E+03 2.78E+03 2 |
| F21 | 7.40E+05 7.73E+05 6 | 1.13E+05 1.27E+05 1 | 9.29E+05 6.55E+05 5 | 4.87E+05 2.88E+05 2 | 5.23E+05 3.45E+05 3 | 5.55E+05 4.00E+05 4 |
| F22 | 6.16E+02 1.54E+02 1 | 7.26E+02 1.21E+02 2 | 9.17E+02 3.09E+02 4 | 9.87E+02 1.88E+02 5 | 8.67E+02 3.04E+02 3 | 1.34E+03 2.83E+02 6 |
| F23 | 3.99E+02 7.66E+01 6 | 3.40E+02 7.70E+00 5 | 3.34E+02 4.88E+00 4 | 3.15E+02 2.00E−02 1 | 3.15E+02 7.00E−02 2 | 3.16E+02 9.60E−01 3 |
| F24 | 3.04E+02 2.66E+01 6 | 2.55E+02 7.98E+00 5 | 2.06E+02 4.44E+00 1 | 2.18E+02 1.12E+01 4 | 2.06E+02 5.36E+00 3 | 2.06E+02 2.38E+00 2 |
| F25 | 2.24E+02 6.67E+00 6 | 2.22E+02 4.31E+00 4 | 2.18E+02 1.88E+01 2 | 2.23E+02 1.63E+01 5 | 2.09E+02 1.31E+01 1 | 2.21E+02 2.08E+01 3 |
| F26 | 1.85E+02 4.57E+01 5 | 1.73E+02 4.93E+01 4 | 1.00E+02 1.10E−01 1 | 1.15E+02 3.76E+01 2 | 1.57E+02 1.13E+02 3 | 1.87E+02 1.08E+02 6 |
| F27 | 9.76E+02 2.08E+02 3 | 9.04E+02 3.26E+02 2 | 1.20E+03 3.32E+02 5 | 1.11E+03 8.11E+01 4 | 7.95E+02 4.61E+02 1 | 1.32E+03 3.90E+02 6 |
| F28 | 2.26E+03 5.47E+02 2 | 3.45E+03 1.67E+03 6 | 2.51E+03 6.25E+02 4 | 1.88E+03 8.69E+02 1 | 2.46E+03 1.21E+03 3 | 2.83E+03 1.38E+03 5 |
| F29 | 3.17E+06 4.31E+06 1 | 2.09E+07 3.46E+07 4 | 5.21E+06 4.84E+06 2 | 7.54E+06 5.27E+06 3 | 3.27E+07 2.74E+07 5 | 3.79E+07 4.11E+07 6 |
| F30 | 2.56E+05 1.79E+05 6 | 5.10E+04 7.03E+04 3 | 9.98E+04 5.98E+04 4 | 3.91E+03 1.91E+03 1 | 8.93E+03 3.30E+03 2 | 1.07E+05 1.76E+05 5 |
| NB/MR | 6/4.3667 | 1/4.2333 | 2/3.6667 | 7/2.8667 | 13/2.0 | 1/4.0667 |

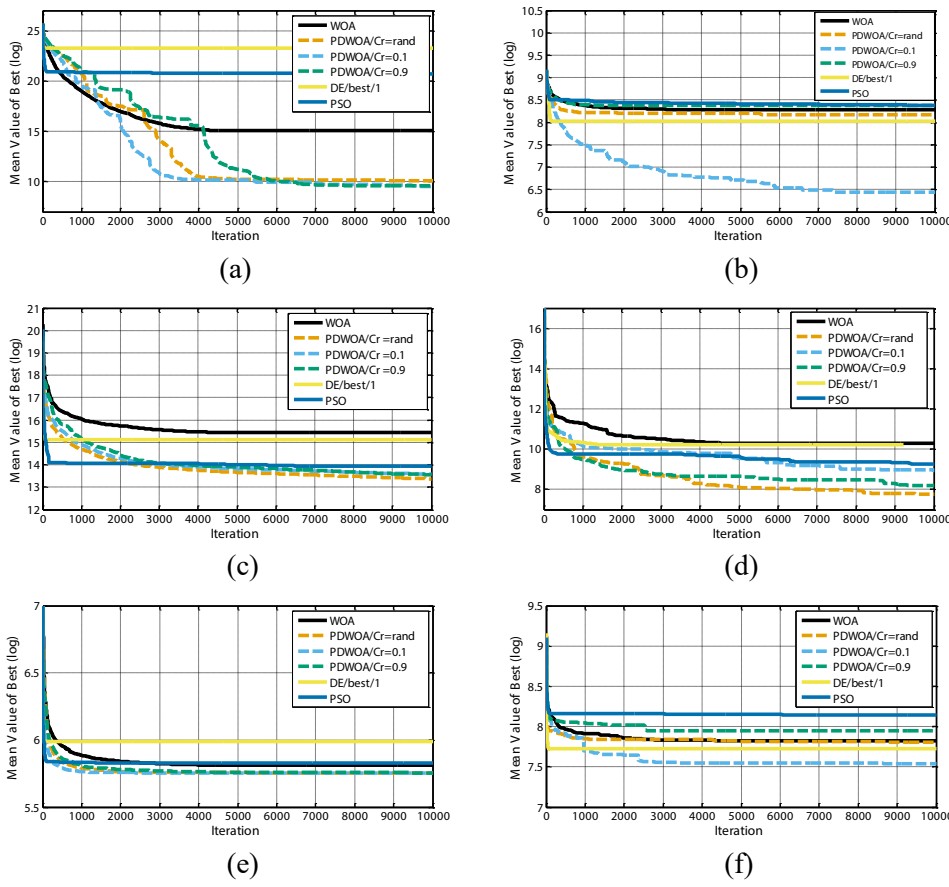

**Figure 3** Convergence curves of optimization algorithms for; (A) F2; (B) F10; (C) F17; (D) F20; (E) F23 and (F) F28 shifted test functions.

to that of the original WOA, PSO, and DE algorithms. It is important to note that, due to the small difference in computational burden between different versions of PDWOA, only the simulation times of PDWOA/Cr = rand are reported in this table. The results indicate that, for 29 and 25 out of 30 test functions, PDWOA has lower mean simulation times than DE and PSO algorithms, respectively. However, for 24 out of 30 test functions, PDWOA has higher mean simulation times than the original WOA. Nonetheless, the maximum increase in mean simulation times by using the proposed improved version of WOA is only about 8%, occurring for test function 1. This increase is not too high considering the degree of improvement in the final solutions.

Table 3 displays a comparative analysis of the performance of the selected variant of the proposed Pbest-guided differential Whale Optimization Algorithm (*i.e.,* PDWOA/Cr = rand) and several other state-of-the-art methods, including Arithmetic Optimization Algorithm (AOA) (*Abualigah et al., 2021*), Hierarchical Multi-swarm Cooperative TLBO (HMCTLBO) (*Zou et al., 2017*), Moth-Flame Optimization algorithm (MFO) (*Mirjalili, 2015*), Adaptive Weighted Particle Swarm Optimizer (AWPSO) (*Liu et al., 2021*), Gaussian bare-bones gradient-based optimization (GOMGBO) (*Qiao et al., 2022*), and Lévy flight

**Table 2  Mean simulation times (s) of 25 runs of different algorithms in solving each of the CEC 2014 test functions.**

| Function | DE/best/1 | PSO | WOA | PDWOA/$Cr$ = rand |
|----------|-----------|-------|-------|-------------------|
| F1 | 6.47 | 5.94 | 4.95 | 5.35 |
| F2 | 5.79 | 5.29 | 4.41 | 4.73 |
| F3 | 5.97 | 5.18 | 4.32 | 4.64 |
| F4 | 5.92 | 5.06 | 4.29 | 4.62 |
| F5 | 5.81 | 5.44 | 4.59 | 4.87 |
| F6 | 30.22 | 29.1 | 30.83 | 30.79 |
| F7 | 5.89 | 5.41 | 4.71 | 5.02 |
| F8 | 5.54 | 5.06 | 4.18 | 4.48 |
| F9 | 5.84 | 5.43 | 4.57 | 4.83 |
| F10 | 6.7 | 6.06 | 5.42 | 5.7 |
| F11 | 7.2 | 6.47 | 5.88 | 5.98 |
| F12 | 10.8 | 10.03 | 9.47 | 9.63 |
| F13 | 5.61 | 4.87 | 4.32 | 4.57 |
| F14 | 5.66 | 5.07 | 4.39 | 4.53 |
| F15 | 6.26 | 5.49 | 4.62 | 4.96 |
| F16 | 5.99 | 5.25 | 4.68 | 4.98 |
| F17 | 6.73 | 5.99 | 5.16 | 5.42 |
| F18 | 6.03 | 5.59 | 4.65 | 4.88 |
| F19 | 11.21 | 9.72 | 9.95 | 9.95 |
| F20 | 6.32 | 5.51 | 4.67 | 4.97 |
| F21 | 6.61 | 5.9 | 5.03 | 5.29 |
| F22 | 6.88 | 6.1 | 5.5 | 5.7 |
| F23 | 12.16 | 11.31 | 10.76 | 11.01 |
| F24 | 10.11 | 9.23 | 8.64 | 8.75 |
| F25 | 11.29 | 10.14 | 9.75 | 9.94 |
| F26 | 40.57 | 36.66 | 39.06 | 38.5 |
| F27 | 38.73 | 36.78 | 38.97 | 38.36 |
| F28 | 14.18 | 12.95 | 12.73 | 12.7 |
| F29 | 15.15 | 13.77 | 14.4 | 13.94 |
| F30 | 11.01 | 10.1 | 9.67 | 9.79 |

Jaya Algorithm (LJA) (*Iacca, dos Santos Junior & Veloso de Melo, 2021*), for solving CEC 2014 test functions.

In this table, the symbols '=', '−', and '+' are used to indicate the comparison between the method under consideration and the proposed PDWOA. The symbol '=' represents an equal result, '−' indicates that the method performs worse than the proposed PDWOA, and '+' signifies that the method performs better than the proposed PDWOA. Furthermore, Nw, Nb, and Ne represent the number of times the considered method performs worse than, better than, or equal to the proposed PDWOA, respectively. The table presents a comprehensive comparison of the results achieved by PDWOA in relation to the benchmarked algorithms, shedding light on the efficacy and competitiveness of PDWOA in addressing the CEC 2014 test functions.

**Table 3 Summary of the results of PDWOA and several state-of-the-art methods for CEC 2014 test functions.**

| Function | AOA Mean Std. -/+/= | HMCTLBO Mean Std. -/+/= | AWPSO Mean Std. -/+/= | GOMGBO Mean Std. -/+/= | MFO Mean Std. -/+/= | LJA Mean Std. -/+/= | PDWOA/*Cr* = rand Mean Std. |
|---|---|---|---|---|---|---|---|
| F1 | 3.026E+07 9.066E+06 − | 2.659E+07 1.490E+07 − | 1.356E+07 1.417E+07 − | 3.238E+07 2.218E+07 − | 7.59E+07 9.77E+07 − | 6.31E+07 1.87E+07 − | 3.15E+06 1.75E+06 |
| F2 | 1.077E+07 1.500E+07 − | 1.332E+06 9.362E+05 − | 6.615E+07 1.141E+08 − | 9.280E+07 1.260E+08 − | 1.36E+10 8.42E+09 − | 4.77E+09 6.03E+08 − | 1.42E+04 1.32E+04 |
| F3 | 2.730E+04 1.121E+04 − | 1.303E+04 1.654E+03 − | 2.669E+04 1.650E+04 − | 1.994E+04 3.669E+03 − | 8.99E+04 4.98E+04 − | 6.91E+04 1.07E+04 − | 3.51E+03 3.56E+03 |
| F4 | 1.887E+02 1.036E+02 − | 2.434E+02 1.483E+02 − | 5.777E+02 5.774E+02 − | 2.207E+02 5.991E+01 − | 1.14E+03 1.13E+03 − | 4.08E+02 5.38E+01 − | 1.23E+02 3.19E+01 |
| F5 | 2.06E+01 5.865E−02 − | 2.084E+01 2.306E−01 − | 2.061E+01 9.313E−02 − | 2.041E+01 1.266E−01 − | 2.04E+01 1.75E −01 − | 2.09E+01 4.97E−02 − | 2.02E+01 1.50E−01 |
| F6 | 4.217E+01 1.739E+00 − | 3.871E+01 2.091E+00 − | 4.103E+01 1.144E+00 − | 4.231E+01 1.126E+00 − | 2.40E+01 3.33E+00 + | 3.39E+01 1.29E+00 − | 2.63E+01 3.33E+00 |
| F7 | 8.071E−01 2.439E−01 − | 6.374E−01 5.306E−01 − | 9.228E−01 2.835E−01 − | 5.078E−01 1.841E−01 − | 1.17E+02 6.91E+01 − | 1.58E+01 2.80E+00 − | 9.00E−02 1.00E−01 |
| F8 | 1.775E+02 2.291E+01 − | 1.979E+02 5.195E+01 − | 2.096E+02 1.096E+01 − | 2.413E+02 6.580E+01 − | 1.43E+02 3.81E+01 − | 2.24E+02 9.93E+00 − | 6.64E+01 1.68E+01 |
| F9 | 2.346E+02 5.240E+01 − | 2.933E+02 1.168E+02 − | 2.188E+02 9.356E+01 − | 1.952E+02 2.288E+01 + | 2.23E+02 6.06E+01 − | 2.61E+02 1.47E+01 − | 2.05E+02 5.24E+01 |
| F10 | 4.937E+03 9.675E+02 − | 5.549E+03 5.726E+02 − | 4.533E+03 9.972E+02 − | 4.475E+03 3.306E+02 − | 3.47E+03 8.85E+02 − | 5.68E+03 3.95E+02 − | 6.27E+02 4.05E+02 |
| F11 | 5.683E+03 5.618E+02 − | 5.416E+03 1.132E+03 − | 4.937E+03 1.579E+03 − | 6.548E+03 9.790E+02 − | 4.15E+03 6.90E+02 − | 6.88E+03 3.12E+02 − | 4.11E+03 8.53E+02 |
| F12 | 2.585E+00 3.545E−02 − | 2.816E+00 4.494E−01 − | 2.124E+00 1.109E−01 − | 1.788E+00 6.116E−01 − | 4.33E −01 2.64E −01 + | 2.49E+00 2.73E–01 − | 1.15E+00 6.70E−01 |
| F13 | 7.508E−01 1.365E−01 − | 5.184E−01 9.286E−02 − | 5.725E−01 3.004E−02 − | 6.601E−01 1.276E−01 − | 2.21E+00 1.34E+00 − | 1.08E+00 1.19E −01 − | 5.10E−01 1.30E−01 |
| F14 | 1.919E−01 2.441E−02 + | 5.586E−01 5.275E−01 − | 2.635E−01 4.831E−02 + | 2.515E−01 1.881E−02 + | 3.54E+01 2.47E+01 − | 4.33E+00 1.70E+00 − | 3.00E−01 5.00E−02 |
| F15 | 1.084E+03 1.878E+02 − | 2.292E+03 1.454E+03 − | 1.119E+03 5.438E+02 − | 3.462E+03 1.196E+03 − | 2.23E+05 5.77E+05 − | 5.05E+01 9.36E+00 − | 4.55E+01 1.21E+01 |

**Table 3** (*continued*)

| Function | AOA Mean Std. -/+/= | HMCTLBO Mean Std. -/+/= | AWPSO Mean Std. -/+/= | GOMGBO Mean Std. -/+/= | MFO Mean Std. -/+/= | LJA Mean Std. -/+/= | PDWOA/$Cr$ = rand Mean Std. |
|---|---|---|---|---|---|---|---|
| F16 | 1.392E+01 6.834E−01 − | 1.262E+01 4.312E−01 − | 1.322E+01 3.863E−01 − | 1.381E+01 9.966E−01 − | 1.27E+01 5.33E −01 − | 1.28E+01 1.78E −01 − | 1.17E+01 4.00E−01 |
| F17 | 1.179E+06 8.704E+05 − | 1.935E+06 1.618E+06 − | 3.150E+06 3.673E+06 − | 1.573E+06 7.309E+05 − | 3.39E+06 4.07E+06 − | 2.63E+06 9.76E+05 − | 7.82E+05 3.34E+05 |
| F18 | 6.615E+03 4.575E+03 − | 8.758E+03 1.307E+04 − | 2.412E+03 2.713E+03 + | 4.646E+03 5.907E+03 − | 5.19E+06 3.61E+07 − | 1.26E+07 1.06E+07 − | 4.34E+03 4.79E+03 |
| F19 | 1.944E+02 3.741E+01 − | 3.112E+02 1.479E+02 − | 1.433E+02 1.725E+01 − | 1.533E+02 6.006E+01 − | 7.36E+01 5.32E+01 − | 3.78E+01 3.45E+01 − | 2.39E+01 2.67E+01 |
| F20 | 1.214E+04 5.535E+03 − | 9.243E+03 7.954E+03 − | 3.171E+04 3.588E+04 − | 1.631E+04 1.192E+04 − | 5.67E+04 4.34E+04 − | 9.92E+03 3.69E+03 − | 7.76E+03 4.60E+03 |
| F21 | 6.694E+05 1.892E+05 − | 9.029E+05 5.565E+05 − | 2.959E+05 1.836E+05 + | 1.346E+06 3.493E+05 − | 7.83E+05 1.18E+06 − | 6.94E+05 2.03E+05 − | 4.87E+05 2.88E+05 |
| F22 | 9.721E+02 1.592E+02 + | 1.358E+03 4.080E+02 − | 1.276E+03 2.694E+02 − | 1.064E+03 1.124E+02 − | 8.67E+02 2.29E+02 + | 5.47E+02 1.05E+02 + | 9.87E+02 1.88E+02 |
| F23 | 2.801E+02 6.936E+01 + | 3.231E+02 4.920E+00 − | 3.208E+02 2.654E+00 − | 3.194E+02 7.377E−01 − | 3.71E+02 3.98E+01 − | 3.43E+02 3.41E+00 − | 3.15E+02 2.00E−02 |
| F24 | 2.094E+02 5.532E+00 + | 2.052E+02 3.837E+00 + | 2.115E+02 2.001E+00 + | 2.065E+02 3.504E+00 + | 2.76E+02 2.73E+01 − | 2.57E+02 4.04E+00 − | 2.18E+02 1.12E+01 |
| F25 | 2.406E+02 3.567E+01 − | 2.293E+02 2.535E+01 − | 2.187E+02 1.509E+01 + | 2.149E+02 2.588E+01 + | 2.14E+02 7.65E+00 + | 2.16E+02 2.58E+00 + | 2.23E+02 1.63E+01 |
| F26 | 1.625E+02 2.418E−01 − | 1.775E+02 7.831E−02 − | 1.669E+02 5.742E+01 − | 3.383E+02 1.202E+02 − | 1.03E+02 1.50E+00 + | 1.01E+02 1.02E −01 + | 1.15E+02 3.76E+01 |
| F27 | 1.259E+03 5.684E+02 − | 1.434E+03 1.252E+02 − | 1.423E+03 7.397E+01 − | 1.501E+03 1.540E+01 − | 9.21E+02 2.23E+02 + | 9.86E+02 2.48E+02 + | 1.11E+03 8.11E+01 |
| F28 | 3.454E+03 5.202E+02 − | 3.811E+03 5.758E+02 − | 3.228E+03 9.435E+02 − | 2.205E+03 1.751E+03 − | 1.12E+03 1.57E+02 + | 1.13E+03 6.63E+01 + | 1.88E+03 8.69E+02 |
| F29 | 7.767E+07 9.097E+07 − | 8.864E+07 7.061E+07 − | 1.572E+08 1.001E+08 − | 1.117E+08 7.572E+07 − | 3.06E+06 3.62E+06 + | 9.82E+05 2.07E+06 + | 7.54E+06 5.27E+06 |
| F30 | 8.012E+04 3.813E+04 − | 9.153E+04 8.882E+04 − | 8.800E+04 1.611E+04 − | 3.903E+05 5.016E+05 − | 5.89E+04 5.40E+04 − | 1.09E+04 4.24E+03 − | 3.91E+03 1.91E+03 |
| Nw/Nb/Ne | 24/4/0 | 29/1/0 | 25/5/0 | 26/4/0 | 22/8/0 | 24/6/0 | |

**Table 4** Average ranking of different algorithms according to the Friedman test.

| Algorithm | RankT | Mean rank |
|---|---|---|
| PDWOA/$Cr$ = rand | 1 | 3.0167 |
| PDWOA/$Cr$ = 0.1 | 2 | 3.9 |
| PSO | 3 | 5.9 |
| PDWOA/$Cr$ = 0.9 | 4 | 5.9667 |
| WOA | 5 | 6.1333 |
| MFO | 6 | 7 |
| LJA | 7 | 7.4333 |
| AOA | 8 | 7.4667 |
| AWPSO | 9 | 7.5333 |
| GOMGBO | 10 | 7.7667 |
| DE/best/1 | 11 | 7.8833 |
| HMCTLBO | 12 | 8 |

## Statistical analysis

In this subsection, we present the results of two non-parametric statistical tests conducted to assess the performance of the proposed improved versions of the whale optimization algorithm (WOA), namely "PDWOA/Cr = rand", "PDWOA/Cr = 0.1', and "PDWOA/Cr = 0.9'. The tests include the Wilcoxon signed-rank test and the Friedman test, which provide insights into the algorithm rankings and pairwise comparisons.

The Friedman test was used to rank the algorithms based on their mean performance across all benchmark functions. The results of the Friedman test are presented in Table 4. In this table, the mean rank index represents the average of the Rank indices of each algorithm for all test functions, while the RankT index shows the rank of each algorithm in the list of sorted mean rank indices. Specifically, PDWOA/Cr = rand achieved the best performance with the lowest mean rank of 3.0167, followed by PDWOA/Cr = 0.1 (mean rank: 3.9), PSO (mean rank: 5.9), and PDWOA/Cr = 0.9 (mean rank: 5.9667).

Additionally, the results of the Wilcoxon signed-rank test with a significance level of 0.05 are presented in Table 5, showing the $p$-values and confidence intervals for pairwise comparisons between PDWOA/Cr = rand and other algorithms. In this table, SoPR and SoNR represent the combined positive and negative ranks. Similarly, MoPR and MoNR represent the average positive and negative ranks, respectively. The notation F(i)<F(j) indicates how many times the first algorithm performs better than the second one, while F(j)<F(i) signifies the opposite scenario. It's important to highlight that in the Wilcoxon test, positive ranks correspond to cases where the first algorithm surpasses the second one. The test results reveal statistically significant differences in performance between PDWOA/Cr = rand and several other algorithms.

PDWOA/Cr = rand was found to have significantly better performance compared to DE/rand/1, PSO, WOA, PDWOA/Cr = 0.9, AOA, AWPSO, GOMGBO, MFO, and LJA, as indicated by the low $p$-values obtained. The confidence intervals further support this finding, showing that PDWOA/Cr = rand consistently outperformed these algorithms over a wide range of objective function values.

**Table 5  The Wilcoxon signed-rank test results between PDWOA/Cr=rand and other algorithms.**

| i | j | MoPR | MoNR | SoPR | SoNR | F(i) <F(j) | F(j) <F(i) | p-value | 0.95 Confidence interval | |
|---|---|---|---|---|---|---|---|---|---|---|
| PDWOA/Cr = rand | DE/rand/1 | 16.261 | 13 | 374 | 91 | 23 | 7 | 2.766e−03 | −126499.9 | −40.05 |
| | PSO | 15.52 | 15.4 | 388 | 77 | 25 | 5 | 8.718e−04 | -23442 | −22.7 |
| | WOA | 16.087 | 10.833 | 370 | 65 | 23 | 6 | 1.014e−03 | −47945.13 | −24.0 |
| | PDWOA/Cr = 0.1 | 15.15 | 14.667 | 303 | 132 | 20 | 9 | 6.607e−02 | −2503.0 | 5.3 |
| | PDWOA/Cr = 0.9 | 14.88 | 15.75 | 372 | 63 | 25 | 4 | 8.685e−04 | −3895.5 | −36.05 |
| | AOA | 16.808 | 7 | 437 | 28 | 26 | 4 | 2.762e−06 | −91195.7 | −86.36 |
| | HMCTLBO | 15.69 | 10 | 455 | 10 | 29 | 1 | 8.009e−08 | −207950.13 | −193.25 |
| | AWPSO | 16.04 | 12.8 | 401 | 64 | 25 | 5 | 2.563e−04 | −42041.75 | −69.45 |
| | GOMGBO | 16.808 | 7 | 437 | 28 | 26 | 4 | 2.762e−06 | −395495.95 | −153.21 |
| | MFO | 16.682 | 12.25 | 367 | 98 | 22 | 8 | 4.665e−03 | -147940 | −25.35 |
| | LJA | 15.375 | 16 | 369 | 96 | 24 | 6 | 4.032e−03 | −103125 | −8.97 |

However, when comparing PDWOA/Cr = rand with PDWOA/Cr = 0.1, the obtained $p$-value (0.0661) suggests that the difference in performance between these two algorithms is not statistically significant at the 0.05 significance level. It is worth noting that PDWOA/Cr = rand still exhibits a slightly better performance trend.

Overall, the results of the statistical tests support the superiority of PDWOA/Cr = rand compared to the other algorithms tested. It demonstrates consistent and competitive performance, as evidenced by its lower mean rank in the Friedman test and its significant performance advantages in the pairwise comparisons based on the Wilcoxon signed-rank test.

## PDWOA for solving constrained engineering optimization

So as to further demonstrate the optimization power of the suggested algorithm, we have selected three renowned engineering problems and solved them with the proposed method. To solve these problems, the population sizes selected for each algorithm is 60 and the number of iterations of each algorithm for each run is 1,000. For each problem, optimization was performed in 30 independent runs. All parameters of the algorithms used here are exactly according to the main references suggested by the algorithm designers.

### Pressure vessel optimal design (engineering problem 1)

The problem is focused on optimally finding two discrete ($x_1$ and $x_2$) and two continuous ($x_3$ and $x_4$) decision variables for the minimization of the cost of a pressure vessel (Fig. 4) subject to three linear and one nonlinear inequality constraint. The optimization variables are the thickness of the shell ($x_1$ or $T_s$), the thickness of the head ($x_2$ or $T_h$), the inner radius ($x_3$ or $R$), and the length of the cylindrical part of the vessel ($x_4$ or $L$) (*Askarzadeh, 2016*).

Minimize:

$$f(X) = 0.6224x_1x_3x_4 + 1.7781x_2x_3^2 + 3.1661x_1^2x_4 + 19.84x_1^2x_3 \qquad (18)$$

subject to:

$$g_1(X) = -x_1 + 0.0193x_3 \leq 0, \qquad (19)$$

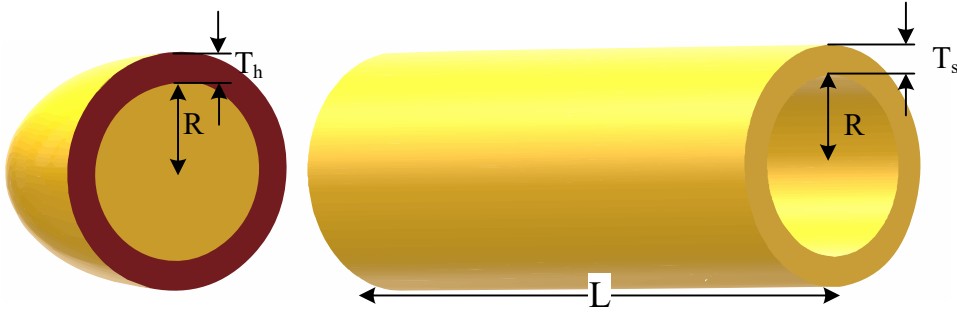

**Figure 4** Schematic of the pressure vessel design problem.

$$g_2(X) = -x_2 + 0.00954x_3 \leq 0, \tag{20}$$

$$g_3(X) = -\pi x_3^2 x_4 - \frac{4}{3}\pi x_3^3 + 1,296,000 \leq 0, \tag{21}$$

$$g_4(X) = x_4 - 240 \leq 0, \tag{22}$$

where $x_1, x_2 \in [0, 100]$, and $x_3, x_4 \in [10, 200]$.

Table 6 presents the results of PDWOA in solving this problem compared to several new algorithms, including quantum-behaved PSO (QPSO) (*Coelho L dos, dos Santos Coelho & Coelho L dos, 2010*), ABC (*Akay & Karaboga, 2012*), GA enhanced with dominance-based tournament selection (GA4) (*Coello Coello et al., 2002*), co-evolutionary PSO (CPSO) (*He & Wang, 2007*), co-evolutionary DE (CDE) (*Huang, Wang & He, 2007*), gaussian quantum-behaved PSO (G-QPSO) (*Coelho L dos, dos Santos Coelho & Coelho L dos, 2010*), Unified PSO (UPSO) (*Parsopoulos & Vrahatis, 2005*), Crow search algorithm (CSA) (*Askarzadeh, 2016*), hybrid GA and artificial immune system (HAIS-GA) (*Coello & Cortés, 2004*), bacterial foraging optimization algorithm (BFOA) (*Mezura-Montes & Hernández-Ocana, 2008*), evolution strategies (ES) (*Mezura-Montes & Coello, 2008*), modified T-Cell Algorithm (*Aragón, Esquivel & Coello, 2010*), GA enhanced with self-adaptive penalty approach (GA3) (*Coello Coello, 2000*), Queuing search (QS) algorithm (*Zhang et al., 2018*), and automatic dynamic penalization (ADP) for GA (BIANCA) (*Montemurro, Vincenti & Vannucci, 2013*), K-means optimizer (KO) (*Minh et al., 2022*), and termite life cycle optimizer (TLCO) (*Minh et al., 2023b; Minh et al., 2023a*). The best solutions for the considered problem found using the original and proposed versions of WOA are presented in Table 7. The results demonstrate the effectiveness of the proposed PDWOA in achieving high-quality solutions for the optimization problem.

### Tension/compression spring optimal design (engineering problem 2)

The problem involves finding three continuous decision variables to minimize the weight of the spring (Fig. 5). The optimization variables are the wire diameter ($d$ or $x_1$), mean

**Table 6  Best statistical results of various algorithms for engineering problem 1.**

| Methods | Best | Mean | Worst | Std. |
|---|---|---|---|---|
| QPSO (*Coelho L dos, dos Santos Coelho & Coelho L dos, 2010*) | 6059.7209 | 6440.3786 | 8017.2816 | 479.2671 |
| ABC (*Akay & Karaboga, 2012*) | 6059.714339 | 6245.308144 | N.A. | 2.05e+02 |
| GA4 (*Coello Coello et al., 2002*) | 6059.9463 | 6177.2533 | 6469.3220 | 130.9297 |
| CPSO (*He & Wang, 2007*) | 6061.0777 | 6147.1332 | 6363.8041 | 86.4545 |
| CDE (*Huang, Wang & He, 2007*) | 6059.7340 | 6085.2303 | 6371.0455 | 43.013 |
| G-QPSO (*Coelho L dos, dos Santos Coelho & Coelho L dos, 2010*) | 6059.7208 | 6440.3786 | 7544.4925 | 448.4711 |
| UPSO (*Parsopoulos & Vrahatis, 2005*) | 6154.70 | 8016.37 | 9387.77 | 745.869 |
| ES (*Mezura-Montes & Coello, 2008*) | 6059.746 | 6850.00 | 7332.87 | 426 |
| T-Cell (*Aragón, Esquivel & Coello, 2010*) | 6390.554 | 6737.065 | 7694.066 | 357 |
| GA3 (*Coello Coello, 2000*) | 6288.7445 | 6293.8432 | 6308.4970 | 7.4133 |
| HAIS-GA (*Coello & Cortés, 2004*) | 6832.584 | 7187.314 | 8012.615 | 276 |
| CSA (*Askarzadeh, 2016*) | 6059.71436343 | 6342.49910551 | 7332.84162110 | 384.94541634 |
| BFOA (*Mezura-Montes & Hernández-Ocana, 2008*) | 6060.460 | 6074.625 | N.A. | 156 |
| BIANCA (*Montemurro, Vincenti & Vannucci, 2013*) | 6059.9384 | 6182.0022 | 6447.3251 | 122.3256 |
| QS (*Zhang et al., 2018*) | 6059.714 | 6060.947 | 6090.526 | N.A. |
| KO | 6059.71475731827 | 6059.72453197228 | N.A. | 0.005942 |
| TLCO | 6059.71433504844 | N.A. | N.A. | N.A. |
| WOA | 6059. 823537 | 6115.250471 | 6314.025148 | 62.35 |
| PDWOA/$Cr = 0.9$ | 6059.714335 | 6064.922632 | 6084.003815 | 12.94 |
| PDWOA/$Cr = $ rand | 6059.714335 | 6063.175326 | 6090.742650 | 24.65 |
| PDWOA/$Cr = 0.1$ | 6059.714335 | 6060.789025 | 6064.324186 | 1.83 |

**Table 7  The best solutions for engineering problem 1.**

| Design variables | WOA | PDWOA/$Cr = 0.9$ | PDWOA/$Cr = $ rand | PDWOA/$Cr = 0.1$ |
|---|---|---|---|---|
| $x_1$ | 0.8125 | 0.8125 | 0.8125 | 0.8125 |
| $x_2$ | 0.4375 | 0.4375 | 0.4375 | 0.4375 |
| $x_3$ | 42.09765834 | 42.09844559 | 42.09844559 | 42.09844559 |
| $x_4$ | 176.6469213 | 176.63659592 | 176.63659592 | 176.63659592 |
| $g_1(X)$ | −1.519403799987718e−05 | −1.130000537585829e−10 | −1.130000537585829e−10 | −1.130000537585829e−10 |
| $g_2(X)$ | −0.0358883394364 | −0.035880829071400 | −0.035880829071400 | −0.035880829071400 |
| $g_3(X)$ | −3.172713808366098 | −2.788752317428589e−05 | −2.788752317428589e−05 | −2.788752317428589e−05 |
| $g_4(X)$ | −63.353078699999998 | −63.363404080000009 | −63.363404080000009 | −63.363404080000009 |
| Best | 6059. 823537 | 6059.714335 | 6059.714335 | 6059.714335 |

coil diameter ($D$ or $x_2$), and the number of active coils ($P$ or $x_3$), subject to one linear and three nonlinear inequality constraints (*Askarzadeh, 2016*).

Minimize:

$$f(X) = (x_3 + 2)x_2 x_1^2 \qquad (23)$$

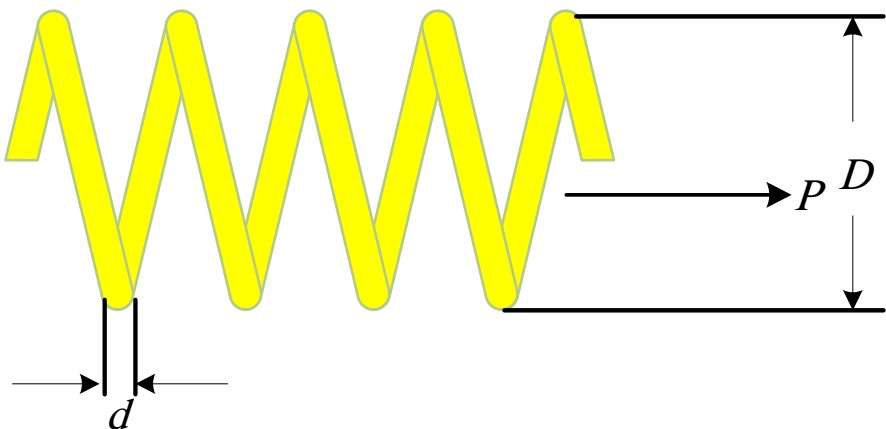

**Figure 5 The tension/compression spring optimal design problem.**

subject to:

$$g_1(X) = 1 - \frac{x_2^3 x_3}{71,785 x_1^4} \le 0, \tag{24}$$

$$g_2(X) = \frac{4x_2^2 - x_1 x_2}{12,566 \left( x_1^3 x_2 - x_1^4 \right)} + \frac{1}{5,108 x_1^2} - 1 \le 0, x_3 \in [2, 15] \tag{25}$$

$$g_3(X) = 1 - \frac{140.45 x_1}{x_2^2 x_3} \le 0, \tag{26}$$

$$g_4(X) = \frac{x_1 + x_2}{1.5} - 1 \le 0. \tag{27}$$

In which $x_1 \in [0.05, 2]$, $x_2 \in [0.25, 1.3]$, and $x_2 \in [2, 15]$.

Table 8 compares the results of the proposed method for solving the engineering problem 2 with several other algorithms, including BFOA (*Mezura-Montes & Hernández-Ocana, 2008*), T-Cell (*Aragón, Esquivel & Coello, 2010*), CDE (*Huang, Wang & He, 2007*), CPSO (*He & Wang, 2007*), a cultural algorithm (CA) (*Coello Coello & Becerra, 2004*), GA4 (*Coello Coello et al., 2002*), GA3 (*Coello Coello, 2000*), TEO (*Kaveh & Dadras, 2017*), G-QPSO (*Coelho L dos, dos Santos Coelho & Coelho L dos, 2010*), SBO (*Ray & Liew, 2003*), evolutionary algorithms ((l + k)-ES) (*Mezura-Montes & Coello, 2005*), UPSO (*Parsopoulos & Vrahatis, 2005*), Grey wolf optimizer (GWO) (*Mirjalili, Mirjalili & Lewis, 2014*), SDO (*Zhao, Wang & Zhang, 2019*), QS (*Zhang et al., 2018*), Water cycle algorithm (WCA) (*Eskandar et al., 2012*), BIANCA (*Montemurro, Vincenti & Vannucci, 2013*), KO (*Minh et al., 2022*), TLCO (*Minh et al., 2023b; Minh et al., 2023a*), planet optimization algorithm (POA) (*Sang-To et al., 2022*), Cuckoo Search Algorithm (CS) (*Cuong-Le et al., 2021*), and the new movement strategy of cuckoo search (NMS-CS) (*Cuong-Le et al., 2021*). Table 9

**Table 8  Best statistical results of various algorithms for engineering problem 2.**

| Methods | Best | Mean | Worst | Std. |
|---|---|---|---|---|
| CA (*Coello Coello & Becerra, 2004*) | 0.012721 | 0.013568 | 0.0151156 | 8.4e−04 |
| BFOA (*Mezura-Montes & Hernández-Ocana, 2008*) | 0.012671 | 0.012759 | N.A. | 1.36e−04 |
| T-Cell (*Aragón, Esquivel & Coello, 2010*) | 0.012665 | 0.012732 | 0.013309 | 9.4e−05 |
| CDE (*Huang, Wang & He, 2007*) | 0.012670 | 0.012703 | 0.012790 | 2.07e−05 |
| CPSO (*He & Wang, 2007*) | 0.0126747 | 0.012730 | 0.012924 | 5.19e−05 |
| TEO (*Kaveh & Dadras, 2017*) | 0.012665 | 0.012685 | 0.012715 | 4.4079e−06 |
| G-QPSO (*Coelho L dos, dos Santos Coelho & Coelho L dos, 2010*) | 0.012665 | 0.013524 | 0.017759 | 1.268e−03 |
| SBO (*Ray & Liew, 2003*) | 0.012669249 | 0.012922669 | 0.016717272 | 5.92e−04 |
| GA4 (*Coello Coello et al., 2002*) | 0.012681 | 0.012742 | 0.012973 | 9.5e−05 |
| GA3 (*Coello Coello, 2000*) | 0.0127048 | 0.012769 | 0.012822 | 3.93e−05 |
| (l + k)-ES (*Mezura-Montes & Coello, 2005*) | 0.012689 | 0.013165 | N.A. | 3.9e−04 |
| UPSO (*Parsopoulos & Vrahatis, 2005*) | 0.01312 | 0.02294 | N.A. | 7.2e−03 |
| GWO (*Mirjalili, Mirjalili & Lewis, 2014*) | 0.0126660 | N.A. | N.A. | N.A. |
| WCA (*Eskandar et al., 2012*) | 0.012665 | 0.012746 | 0.012952 | 8.06e−05 |
| BIANCA (*Montemurro, Vincenti & Vannucci, 2013*) | 0.012671 | 0.012681 | 0.012913 | 5.1232e−05 |
| SDO (*Zhao, Wang & Zhang, 2019*) | 0.0126663 | 0.0126724 | 0.0126828 | 6.1899e−06 |
| QS (*Zhang et al., 2018*) | 0.012665 | 0.012666 | 0.012669 | N.A. |
| KO | 0.012665994 | 0.012917292 | N.A. | 0.00030139 |
| TLCO | 0.0126652328 | N.A. | N.A. | N.A. |
| POA | 0.01266588 | N.A. | N.A. | N.A. |
| CS | 0.012665871 | N.A. | N.A. | N.A. |
| NMS-CS | 0.012665233 | N.A. | N.A. | N.A. |
| WOA | 0.012667 | 0.013586 | 0.018416 | 5.05e−03 |
| PDWOA/$Cr = 0.9$ | 0.012665 | 0.012695 | 0.012842 | 8.75e−06 |
| PDWOA/$Cr = $ rand | 0.012665 | 0.012706 | 0.012907 | 1.18e−05 |
| PDWOA/$Cr = 0.1$ | 0.012665 | 0.012665 | 0.012666 | 9.22e−08 |

**Table 9  The best solutions for engineering problem 2.**

| Design variables | WOA | PDWOA/$Cr = 0.9$ | PDWOA/$Cr = $ rand | PDWOA/$Cr = 0.1$ |
|---|---|---|---|---|
| $x_1$ | 0.0517315934 | 0.0515902788 | 0.0516488707 | 0.0516911532 |
| $x_2$ | 0.3577231396 | 0.3543444741 | 0.3557507777 | 0.3567674033 |
| $x_3$ | 11.2318481822 | 11.4295493851 | 11.3460406066 | 11.2862994555 |
| $g_1(X)$ | −8.105263015556474e−05 | −2.067013876949631e−06 | −1.124305991995200e−05 | −1.953083625561014e−05 |
| $g_2(X)$ | −4.202663105634663e−05 | −3.336199576375876e−06 | −1.936754291387288e−06 | −1.509602815197297e−06 |
| $g_3(X)$ | −4.055129747949155 | −4.049044297898749 | −4.051804364878148 | −4.053776839282882 |
| $g_4(X)$ | −0.727030178 | −0.729376831400 | −0.7284002344 | −0.727694295666667 |
| Best | 0.012667 | 0.012665 | 0.012665 | 0.012665 |

provides the best solutions obtained by the original and proposed versions of WOA for this problem. The findings indicate that the suggested PDWOA is successful in attaining excellent solutions for the optimization issue.

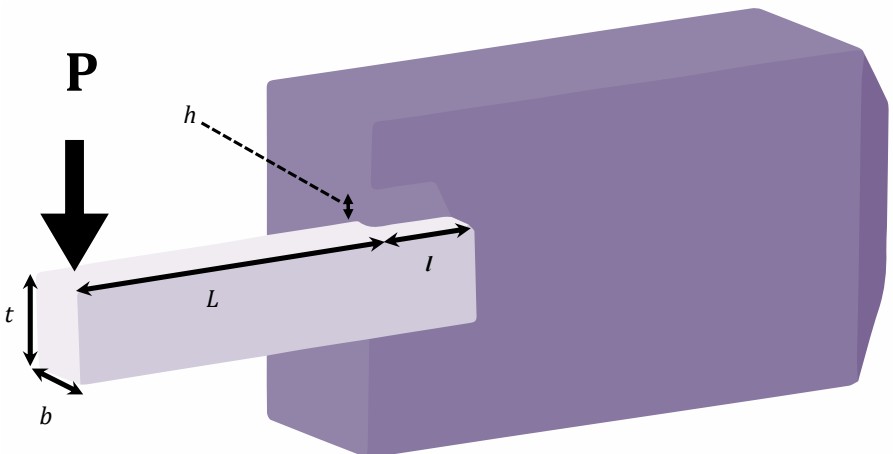

**Figure 6** Schematic of welded beam optimal design problem.

### Welded beam optimal design (engineering problem 3)

The problem is focused on optimally finding four continuous decision variables for minimizing the cost of a welded beam (Fig. 6) subject to two linear and five nonlinear inequality constraints. The optimization variables are $x_1$ or h, $x_2$ or $l$, $x_3$ or $t$, and $x_4$ or $b$ (*Askarzadeh, 2016*).

Minimize:

$$f(X) = 1.10471x_2x_1^2 + 0.04811x_3x_4(14 + x_2) \tag{28}$$

subject to:

$$g_1(X) = \tau(x) - \tau_{\max} \leq 0, \tag{29}$$

$$g_2(X) = \sigma(x) - \sigma_{\max} \leq 0, \tag{30}$$

$$g_3(X) = x_1 - x_4 \leq 0, \tag{31}$$

$$g_4(X) = 0.10471x_1^2 + 0.04811x_3x_4(14 + x_2) - 5 \leq 0. \tag{32}$$

$$g_5(X) = 0.125 - x_1 \leq 0, \tag{33}$$

$$g_6(X) = \delta(x) - \delta_{\max} \leq 0, \tag{34}$$

$$g_7(X) = P - P_c(x) \leq 0, \tag{35}$$

$$\tau(x) = \sqrt{(\tau')^2 + 2\tau'\tau''\frac{x_2}{2R} + (\tau'')^2} \qquad (36)$$

$$\tau' = \frac{P}{\sqrt{2}x_1x_2}, \tau'' = \frac{MR}{J}, \qquad (37\text{-}38)$$

$$M = P\left(L + \frac{x_2}{2}\right), R = \sqrt{\frac{x_2^2}{4} + \left(\frac{x_1+x_3}{2}\right)^2}, \delta(x) = \frac{4PL^3}{Ex_3^3x_4} \qquad (39\text{-}41)$$

$$J = 2\left[\sqrt{2}x_1x_2\left\{\frac{x_2^2}{12} + \left(\frac{x_1+x_3}{2}\right)^2\right\}\right], \sigma(x) = \frac{6PL}{x_4x_3^2}, \qquad (42\text{-}43)$$

$$P_c(x) = \frac{4.013E\sqrt{\frac{x_4^6x_3^2}{36}}}{L^2}\left(1 - \frac{x_3}{2L}\sqrt{\frac{E}{4G}}\right), \qquad (44)$$

where $P = 6{,}000$ lb; $L = 14$ in; $E = 30e6$ psi; $G = 12e6$ psi; $\tau_{max} = 13{,}000$ psi; $\sigma_{max} = 30{,}000$ psi; $\delta_{max} = 0.25$ in; $x_1 \in [0.1, 2]$; $x_2 \in [0.1, 10]$; $x_3 \in [0.1, 10]$; and $x_4 \in [0.1, 2]$.

Table 10 presents the results of the proposed method for solving the engineering problem 3 in comparison to several other algorithms, including a cooperative PSO with stochastic movements (EPSO) (*Ngo, Sadollah & Kim, 2016*), BFOA (*Mezura-Montes & Hernández-Ocana, 2008*), T-Cell Algorithm (*Aragón, Esquivel & Coello, 2010*), CDE (*Huang, Wang & He, 2007*), CPSO (*He & Wang, 2007*), Derivative-Free Filter Simulated Annealing Method (FSA) (*Hedar & Fukushima, 2006*), TEO (*Kaveh & Dadras, 2017*), SBO (*Ray & Liew, 2003*), GA4 (*Coello Coello et al., 2002*), (l + k)-ES (*Mezura-Montes & Coello, 2005*), UPSO (*Parsopoulos & Vrahatis, 2005*), GWO (*Mirjalili, Mirjalili & Lewis, 2014*), SFO (*Shadravan, Naji & Bardsiri, 2019*), HGSO (*Hashim et al., 2019*), WCA (*Eskandar et al., 2012*), BIANCA (*Montemurro, Vincenti & Vannucci, 2013*), SBO (*Ray & Liew, 2003*), KO (*Minh et al., 2022*), TLCO (*Minh et al., 2023b; Minh et al., 2023a*), POA (*Sang-To et al., 2022*), CS (*Cuong-Le et al., 2021*), and NMS-CS (*Cuong-Le et al., 2021*). Table 11 presents the best solutions found by the original and proposed versions of WOA for this problem. The outcomes exhibit the efficacy of the suggested PDWOA in attaining top-notch solutions for the optimization issue.

## DISCUSSION AND FUTURE STUDIES

Table 12 presents the best solutions found by the proposed and original versions of WOA. The results indicate that optimizing the algorithm's parameters, particularly the value of $Cr$, can significantly enhance the optimization performance. For example, when comparing the final results for F29 (shown in Table 1), the original WOA yielded the best mean value

**Table 10  Best statistical results of various algorithms for engineering problem 3.**

| Methods | Best | Mean | Worst | Std. |
|---|---|---|---|---|
| EPSO (*Ngo, Sadollah & Kim, 2016*) | 1.7248530 | 1.7282190 | 1.7472200 | 5.62e−03 |
| BFOA (*Mezura-Montes & Hernández-Ocana, 2008*) | 2.3868 | 2.4040 | N.A. | 1.6e−02 |
| T-Cell (*Aragón, Esquivel & Coello, 2010*) | 2.3811 | 2.4398 | 2.7104 | 9.314e−02 |
| CDE (*Huang, Wang & He, 2007*) | 1.73346 | 1.768158 | 1.824105 | 2.2194e−02 |
| CPSO (*He & Wang, 2007*) | 1.728024 | 1.748831 | 1.782143 | 1.2926e−02 |
| FSA (*Hedar & Fukushima, 2006*) | 2.3811 | 2.4041 | 2.4889 | N.A. |
| TEO (*Kaveh & Dadras, 2017*) | 1.725284 | 1.768040 | 1.931161 | 5.81661e−02 |
| SBO (*Ray & Liew, 2003*) | 2.3854347 | 3.0025883 | 6.3996785 | 9.59e−01 |
| GA4 (*Coello Coello et al., 2002*) | 1.728226 | 1.792654 | 1.993408 | 7.47e−02 |
| (l + k)-ES (*Mezura-Montes & Coello, 2005*) | 1.724852 | 1.777692 | N.A. | 8.8e−02 |
| UPSO (*Parsopoulos & Vrahatis, 2005*) | 1.92199 | 2.83721 | N.A. | 6.83e−01 |
| GWO (*Mirjalili, Mirjalili & Lewis, 2014*) | 1.72624 | N.A. | N.A. | N.A. |
| SFO (*Shadravan, Naji & Bardsiri, 2019*) | 1.73231 | N.A. | N.A. | N.A. |
| HGSO (*Hashim et al., 2019*) | 1.7260 | 1.7265 | 1.7325 | 7.66e−03 |
| WCA (*Eskandar et al., 2012*) | 1.724856 | 1.726427 | 1.744697 | 4.29e−03 |
| BIANCA (*Montemurro, Vincenti & Vannucci, 2013*) | 1.725436 | 1.752201 | 1.793233 | 2.3001e−02 |
| KO | 1.725344872 | 1.75727933 | N.A. | 0.029125 |
| TLCO | 1.724852433 | N.A. | N.A. | N.A. |
| POA | 1.72564 | N.A. | N.A. | N.A. |
| CS | 1.73139841 | N.A. | N.A. | N.A. |
| NMS-CS | 1.72620872 | N.A. | N.A. | N.A. |
| WOA | 1.7273929 | 2.2852435 | 3.2784166 | 2.62 |
| PDWOA/$Cr = 0.9$ | 1.7248523 | 1.7310629 | 1.7491305 | 9.83e−04 |
| PDWOA/$Cr = $ rand | 1.7248523 | 1.7588375 | 1.7709064 | 2.06e−03 |
| PDWOA/$Cr = 0.1$ | 1.7248523 | 1.7259521 | 1.7340485 | 5.93e−05 |

**Table 11  The best solutions for engineering problem 3.**

| Design variables | WOA | PDWOA/$Cr = 0.9$ | PDWOA/$Cr = $ rand | PDWOA/$Cr = 0.1$ |
|---|---|---|---|---|
| $x_1$ | 0.2053718352 | 0.2057296398 | 0.20572963980 | 0.20572963980 |
| $x_2$ | 3.4771582193 | 3.470488670 | 3.4704886655 | 3.4704886655 |
| $x_3$ | 9.0472014495 | 9.0366239108 | 9.0366239101 | 9.0366239101 |
| $x_4$ | 0.2057779509 | 0.2057296398 | 0.2057296398 | 0.2057296398 |
| $g_1(X)$ | −9.453362733127506 | −1.514258474344388e−05 | −2.265333023387939e−07 | −2.265333023387939e−07 |
| $g_2(X)$ | −77.134747837790201 | −4.967081622453407e−06 | −3.193272277712822e−07 | −3.193272277712822e−07 |
| $g_3(X)$ | −4.061157000000149e−04 | 0.0 | 0.0 | 0.0 |
| $g_4(X)$ | −3.43020541215535 | −3.432983784788387 | −3.432983785311915 | −3.432983785311915 |
| $g_5(X)$ | −0.08037183520 | −0.08072963980 | −0.080729639800 | −0.080729639800 |
| $g_6(X)$ | −0.235594362774535 | −0.235540322587856 | −0.235540322584496 | −0.235540322584496 |
| $g_7(X)$ | −8.845720840467948 | −1.411061930411961e−06 | −1.105492628994398e−06 | −1.105492628994398e−06 |
| Best | 1.7273929 | 1.7248523 | 1.7248523 | 1.7248523 |

**Table 12  Summary of the best results for CEC 2014 test functions for WOA algorithms.**

| Function | WOA<br>Best | PDWOA/ $Cr = $ rand<br>Best | PDWOA/ $Cr = 0.1$<br>Best | PDWOA/ $Cr = 0.9$<br>Best |
|---|---|---|---|---|
| F1 | 19833281.23 | 680089.84 | 1348332.9 | 1288387.42 |
| F2 | 1994653.36 | 1105.87 | 1.24 | 93.08 |
| F3 | 15105.52 | 101.54 | 48.61 | 1950.79 |
| F4 | 132.74 | 67.85 | 78.74 | 73.05 |
| F5 | 20.08 | 20 | 20 | 19.99 |
| F6 | 31.78 | 28.38 | 21.54 | 37.93 |
| F7 | 0.97 | 1.94e−5 | 2.55e−6 | 1.28e−3 |
| F8 | 137.4 | 156.99 | 38.8 | 128.35 |
| F9 | 176.36 | 143.27 | 138.3 | 190.04 |
| F10 | 2543.45 | 2428.22 | 44.13 | 3793.69 |
| F11 | 3809.73 | 3456.96 | 2876.84 | 2874.24 |
| F12 | 0.43 | 0.69 | 0.18 | 1.17 |
| F13 | 0.4 | 0.45 | 0.39 | 0.44 |
| F14 | 0.21 | 0.23 | 0.22 | 0.25 |
| F15 | 50.75 | 191.85 | 22.36 | 375.04 |
| F16 | 11.81 | 12.65 | 11.14 | 12.12 |
| F17 | 2514752.92 | 211439.89 | 477893.91 | 420326.09 |
| F18 | 483.1 | 270.89 | 92.49 | 215.83 |
| F19 | 24.86 | 17.97 | 10.33 | 36.43 |
| F20 | 8808.02 | 350.79 | 635.06 | 958.96 |
| F21 | 290044.26 | 126426.49 | 213375.89 | 44587.71 |
| F22 | 451.24 | 446.64 | 580.79 | 896.22 |
| F23 | 324.86 | 315.26 | 315.25 | 315.3 |
| F24 | 201.88 | 201.38 | 202.81 | 202.16 |
| F25 | 200 | 200 | 205.68 | 200 |
| F26 | 100.35 | 100.57 | 100.23 | 100.49 |
| F27 | 459.35 | 409.61 | 1015.92 | 444.06 |
| F28 | 1576.31 | 200 | 200 | 200 |
| F29 | 38445.29 | 1364.48 | 1141.58 | 1123.79 |
| F30 | 52816 | 5011.8 | 2201.07 | 5668.48 |

with a slight difference, whereas the proposed algorithm with $Cr$ equal to 0.1 achieved the best final value, as demonstrated in Table 12. As part of future work, an efficient modification can be explored to improve the Mean obtained by PDWOA and align it with the best value.

Figure 7 illustrates the average performance of the suggested algorithm across multiple test functions in 30 runs. These results demonstrate the significant improvement achieved by the suggested algorithm in enhancing WOA. Although the suggested algorithm is robust and effective in many cases, further development can be pursued by exploring numerous enhanced versions of PSO or DE. In future work, we will present some examples of these enhanced versions. For instance, we can draw inspiration from the colonial competitive differential evolution algorithm (*Ghasemi et al., 2016*), which suggests distributing the

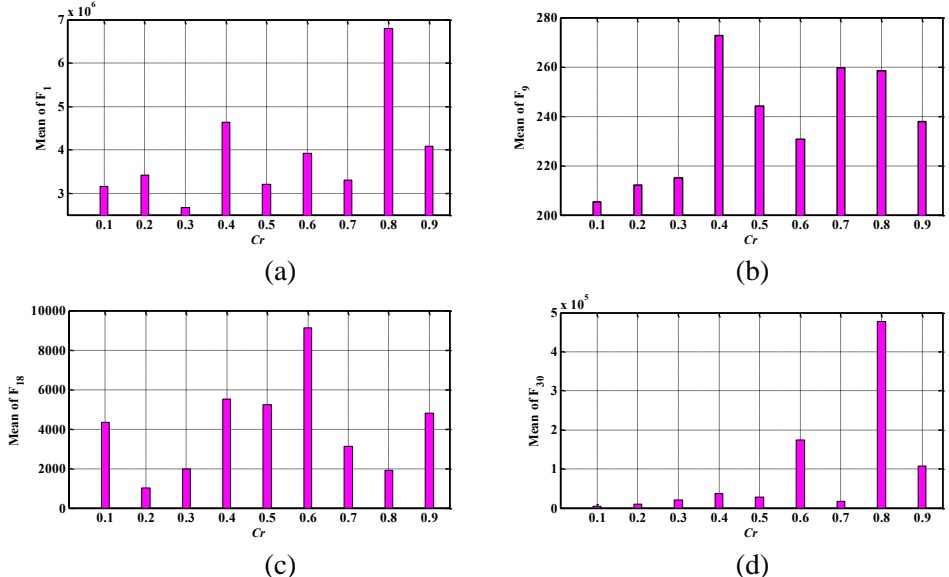

**Figure 7** **The mean index obtained for different test functions by PDWOA having different Cr values; (A) F1 (unimodal); (B) F9 (simple multimodal); (C) F18 (hybrid); (D) F30 (composition).**

population into several groups and conducting colonial competition between these groups using a specific mutation for each group. To enhance the proposed version, we can implement a similar mechanism by dividing the WOA population into multiple groups, where the best member of each group serves as the leader, and conduct colonial competition among these groups.

Additionally, instead of using the specific mutation equation defined in Eq. (17), we can employ alternative mutations (or crossover coefficients) for each group of whales. By leveraging the efficient operators from various evolutionary algorithms, we can increase the population diversity during iterations. This approach, guided by multiple leaders within distinct groups, allows the population to explore several different areas in the search space, effectively avoiding local optima. In this context, several new optimization algorithms that involve population division into multiple groups (*Mallipeddi et al., 2011*; *Zhang, 2015*; *Chen et al., 2018*; *Band et al., 2022*) can be applied to enhance the proposed version of WOA.

Furthermore, a highly effective and adaptive method for selecting *Cr* was proposed in *Zhang & Sanderson (2009)*, which is recognized as one of the most powerful versions of DEs. The mutation equation presented in Eq. (17) has drawn inspiration from this method. In future studies, we can explore the application of this strategy to enhance the performance of the suggested method. Additionally, other efficient adaptive techniques proposed in *Brest et al. (2006)* and *Zhu et al. (2013)* can be further investigated as potential avenues to improve the proposed algorithm. Additionally, there are several new models of DE proposed in the literature that extend beyond the scope of this study but warrant further investigation. These models include fuzzy adaptive differential evolution

(*Al-Dabbagh et al., 2014*), Gaussian bare-bones differential evolution (*Wang et al., 2013*), and parallel DE with self-adapting control parameters and generalized opposition-based learning (*Wang, Rahnamayan & Wu, 2013*). Future studies can delve into these models in more detail to explore their potential implications.

## CONCLUSIONS

In this study, we addressed the limitations of the original WOA, such as susceptibility to getting trapped in locally optimal solutions, particularly in complex real-world problems. To overcome these drawbacks, we proposed a new and high-performing version of WOA called PDWOA. The performance of PDWOA was evaluated by comparing it with the original WOA on 30 shifted test functions from CEC2014, each with a dimension of 30, under identical conditions. The simulation results demonstrated the efficiency of the suggested algorithm in achieving optimal solutions for the test cases. Moreover, the proposed PDWOA algorithm was evaluated using two non-parametric statistical tests, including the Friedman test and the Wilcoxon signed-rank test, which confirmed its superior performance compared to other algorithms. Furthermore, PDWOA was applied to three real-world engineering problems, providing additional evidence of its optimization performance. Additionally, we discussed models of powerful algorithms from the literature that could be explored in future studies to further enhance the proposed algorithm. By integrating these models into the proposed formulation, we aim to achieve accurate solutions for a wider range of real-world optimization problems.

### Funding

The research was supported by the Excellence Project of Faculty of Science, University of Hradec Králové, Czech Republic, 2210/2023-2024. The funders had no role in study design, data collection and analysis, decision to publish, or preparation of the manuscript.

### Grant Disclosures

The following grant information was disclosed by the authors:
The Excellence Project of Faculty of Science, University of Hradec Králové, Czech Republic: 2210/2023-2024.

### Competing Interests

The authors declare there are no competing interests.

### Author Contributions

- Abolfazl Rahimnejad performed the experiments, analyzed the data, performed the computation work, prepared figures and/or tables, authored or reviewed drafts of the article, and approved the final draft.
- Ebrahim Akbari conceived and designed the experiments, performed the experiments, analyzed the data, performed the computation work, prepared figures and/or tables, authored or reviewed drafts of the article, and approved the final draft.

- Seyedali Mirjalili analyzed the data, authored or reviewed drafts of the article, and approved the final draft.
- Stephen Andrew Gadsden analyzed the data, authored or reviewed drafts of the article, and approved the final draft.
- Pavel Trojovský analyzed the data, authored or reviewed drafts of the article, and approved the final draft.
- Eva Trojovská analyzed the data, authored or reviewed drafts of the article, and approved the final draft.

## Data Availability

The data is available at Zenodo: Ebrahim Akbari, & Abolfazl Rahimnejad. (2023). PDWOA MATLAB Code. Zenodo. https://doi.org/10.5281/zenodo.7608100.

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
