# Peer review of "An improved hybrid whale optimization algorithm for global optimization and engineering design problems"

_PeerJ Computer Science, doi:10.7717/peerj-cs.1557_

## Round 0.1 · original submission · Major Revisions

Dear authors,

The reviews for your manuscript are included at the bottom of this letter. We ask that you make changes to your manuscript based on those comments. Thank you.

·

Basic reporting

no comment

Experimental design

no comment

Validity of the findings

All results shoud be compared to other recently algorithms.

Additional comments

The manuscript presents a new version of WOA based on the movement strategies of PSO and DE. Some benchmark functions and engineering problems are used to check the reliability and the effectiveness of the proposed algorithm. The ideal using hybrid technique is normal and not reach high academic. However, the following issues should be addressed before the reviewer recommends it to be published in the Journal.
1/ Optimization algorithm have proved the effectiveness in engineering field. Recently, many effective algorithms have been proposed such as: TLCO https://doi.org/10.1016/j.eswa.2022.119211; KO https://doi.org/10.1016/j.knosys.2022.109189); POA https://doi.org/10.1038/s41598-022-12030-w; NMS-CS https://doi.org/10.1016/j.eswa.2021.115669
2/ WOA maybe the same metaphor with GWO. Both two algorithms were developed by the owner. The author should explain in more detail how difference between two algorithms.
3/ The hybrid technique is sure to enhance the effectiveness however, it will increase the computational cost. The authors must compare the time for each benchmark functions with the other algorithms.
4/ nowadays, there are many optimization algorithms have been proposed even everyday.
I wonder Why the authors need to proposed a new version, some recent algorithms can handle well in solving optimization problems.
5/ There are many optimization algorithms proposed, why the authors use WOA optimization algorithm? To verify the effectiveness of the proposed method, the authors should compare the performance of GA with some optimization algorithms (TLCO and KO). The lack of any studies in this respect in the study is a major shortcoming. The reviewer suggests to add this section.

Reviewer 2 ·

Basic reporting

- In Abstract, no information is given regarding the number and characteristics or names of benchmark and real-world problems used for testing. Numerical information about the results obtained is not given.
- Keywords are not given in alphabetical order.
- It is recommended to shorten or split sentences for clarity and readability.
- There are many metaheuristic algorithms and their improved versions in the literature. Specifically, why a new hybrid version of WOA is needed is not stated. Considering that there are too many metaheuristic algorithms in the literature, don't these newly proposed hybrid algorithms, which will increase the options too much, harm the researchers working in this field about which algorithm to choose from among so many options?
- It is suggested that the main contributions of the study to science should be given in items.
- Due to the absences of section numbering, organization of the paper written as the last paragraph of the Introduction section should be rewritten.
- Some paragraphs are too long for readability. They can be divided into two or more.
- In the literature review, studies in which WOA was used as an interesting solution method in the sentiment classification problem were ignored.
- Equations should be referenced using the corresponding equation number. "... as follows:", "... as:" etc. form should not be used.
- Equations in texts should be italicized as in formulas.
- The expression "Where (7) and (8) denotes the encircling prey phase, (11) and (12) models the search for prey" in line 257 should be corrected as "Where (7) and (8) denote the encircling prey phase, (11) and (12) model the search for prey".
- "." the sign is used both for decimal notation and for multiplication sign operation.

Experimental design

- The literature is summarized in a crude form. The defiance’s and research gaps in the given literature are not specified.
- In the "DE-based Modification" sub-title where the proposed method is explained, Fig1 and Fig2, which give the results of the study, should not be mentioned.
- Constrained handling methods for real engineering design problems are not described.
- It is not clear on what basis the other methods selected for comparison were chosen. The reason for choosing different algorithms in the comparison for engineering problems is not explained.

Validity of the findings

- Statistical test results are not presented.
- In Conclusion, why did the authors use the concept of "efficient" for the algorithm. "efficient" has a different meaning in algorithm analysis.

Additional comments

The authors proposed a new method, "Pbest-guided Differential WOA", which they hybridized with PSO and DE in order to eliminate the weakness of WOA they identified in this study. My suggestions for the article are listed below:
- In Abstract, no information is given regarding the number and characteristics or names of benchmark and real-world problems used for testing. Numerical information about the results obtained is not given.
- Keywords are not given in alphabetical order.
- It is recommended to shorten or split sentences for clarity and readability.
- There are many metaheuristic algorithms and their improved versions in the literature. Specifically, why a new hybrid version of WOA is needed is not stated. Considering that there are too many metaheuristic algorithms in the literature, don't these newly proposed hybrid algorithms, which will increase the options too much, harm the researchers working in this field about which algorithm to choose from among so many options?
- It is suggested that the main contributions of the study to science should be given in items.
- Due to the absences of section numbering, organization of the paper written as the last paragraph of the Introduction section should be rewritten.
- Some paragraphs are too long for readability. They can be divided into two or more.
- The literature is summarized in a crude form. The defiance’s and research gaps in the given literature are not specified.
- In the literature review, studies in which WOA was used as an interesting solution method in the sentiment classification problem were ignored.
- Equations should be referenced using the corresponding equation number. "... as follows:", "... as:" etc. form should not be used.
- Equations in texts should be italicized as in formulas.
- The expression "Where (7) and (8) denotes the encircling prey phase, (11) and (12) models the search for prey" in line 257 should be corrected as "Where (7) and (8) denote the encircling prey phase, (11) and (12) model the search for prey".
- In the "DE-based Modification" sub-title where the proposed method is explained, Fig1 and Fig2, which give the results of the study, should not be mentioned.
- Constrained handling methods for real engineering design problems are not described.
- "." the sign is used both for decimal notation and for multiplication sign operation.
- It is not clear on what basis the other methods selected for comparison were chosen. The reason for choosing different algorithms in the comparison for engineering problems is not explained.
- In Conclusion, why did the authors use the concept of "efficient" for the algorithm. "efficient" has a different meaning in algorithm analysis.
- Statistical test results are not presented.

Reviewer 3 ·

Basic reporting

This manuscript presents a study on using titanium alloy welded joints' fatigue data as analysis data and using the neighborhood rough set reduction with improved firefly algorithm efficient method of fitting stress-life curves. The authors used the continuous iteration of the firefly algorithm as the search strategy and adopted the neighborhood rough set to decrease attributes. The paper can be accepted for publication after considering the following major revision.
1- The Scientific merit and novelty of the article are not clear. The authors should explain clearly in the abstract what is the novelty of the proposed method and what is the added value in this article?
2- The abstract should be improved and include clear statements about objectives, methodology and findings.
3- There are many typos; see for example on page 1 ‘It mainly consist of the local stress-strain method ‘ and on page 2 ‘At present, the master S-N curve method based on equivalent structural stress has been widely uesd.’ The authors should check typos throughout the MS.
4- The authors should justify the choice of improved firefly algorithm; there are many other optimization algorithms; see the references below. Why not use other algorithms and compare for the best performance?
5- The literature should be enhanced by referring to recent works on optimization techniques and their applications, e.g. Advances in Engineering Software 176, 103399(2023); Thin-Walled Structures 182, Part B, 110267 (2023).
6- Conclusion should be more carefully rewritten, summarizing what has been learned and why it is interesting and useful.
7- English style should also be improved.

Experimental design

.

Validity of the findings

.

Additional comments

.

---

## Round 0.2 · accepted · Accept

Dear authors,

Two of the reviewers did not agree to re-review this revision but I have checked the revision and the response letter. Thank you for clearly addressing all of the reviewers' comments. Your article is accepted for publication now.

Best wishes,

·

Basic reporting

'no comment'

Experimental design

'no comment'

Validity of the findings

'no comment'

Additional comments

The paper was revised. The paper is suitable for publication.